# Toward Generalizing Visual Brain Decoding to Unseen Subjects

**Xiangtao Kong** [1,*] **Kexin Huang** [1,*] **, Ping Li** [1,†] **Lei Zhang** [1,†]
[1]The Hong Kong Polytechnic University
{xiangtao.kong, kexin0.huang}@connect.polyu.hk,
ping2.li@polyu.edu.hk, cslzhang@comp.polyu.edu.hk

## ABSTRACT

Visual brain decoding aims to decode visual information from human brain activities. Despite the great progress, one critical limitation of current brain decoding research lies in the lack of generalization capability to unseen subjects. Prior work typically focuses on decoding brain activity of individuals based on the observation that different subjects exhibit different brain activities, while it remains unclear whether brain decoding can be generalized to unseen subjects. This study aims to answer this question. We first consolidate an image-fMRI dataset consisting of stimulus-image and fMRI-response pairs, involving 177 subjects in the movie-viewing task of the Human Connectome Project (HCP). This dataset allows us to investigate the brain decoding performance with the increase of participants. We then present a learning paradigm that applies uniform processing across all subjects, instead of employing different network heads or tokenizers for individuals as in previous methods, so that we can accommodate a large number of subjects to explore the generalization capability across different subjects. A series of experiments are conducted and we have the following findings. First, the network exhibits clear generalization capabilities with the increase of training subjects. Second, the generalization capability is common to popular network architectures (MLP, CNN and Transformer). Third, the generalization performance is affected by the similarity between subjects. Our findings reveal the inherent similarities in brain activities across individuals. With the emergence of larger and more comprehensive datasets, it is possible to train a brain decoding foundation model in the future. Codes and models can be found at https://github.com/Xiangtaokong/TGBD.

## 1 INTRODUCTION

Visual brain decoding (Kay et al., 2008; Kamitani & Tong, 2005; Naselaris et al., 2011) aims to decode visual information from human brain activities, including tasks of brain-image classification (Kaur & Gandhi, 2019; Zhou et al., 2024), retrieval (Scotti et al., 2024a; Xia et al., 2024) and reconstruction (Takagi & Nishimoto, 2023; Ozcelik & VanRullen, 2023; Ferrante et al., 2024; Scotti et al., 2024a), and so on. It involves analyzing neural patterns collected via brain imaging techniques like functional magnetic resonance imaging (fMRI) (Schirrmeister et al., 2017; Benchetrit et al., 2023; Kamitani & Tong, 2005) or electroencephalography (EEG) (Schirrmeister et al., 2017; Vallabhaneni et al., 2021) to infer the visual information received by the participants. Among them, fMRI is favored by researchers because of its high spatial resolution and more informative depiction of the whole brain activity, which has resulted in a number of important decoding studies (Allen et al., 2022; Takagi & Nishimoto, 2023; Scotti et al., 2024a) with the help of deep learning techniques.

A major limitation of current brain decoding research, however, lies in the lack of generalization capability to unseen subjects. That is, the trained decoding models can hardly be applied to new, unseen individuals not in the original training. Such a limitation can be owed to two reasons. First, there are individual differences of the brain activities across subjects (Haxby et al., 2020). Therefore, it is assumed that brain decoding cannot be generalized and hence researchers are focused on

---

*Equal contributions.
†Corresponding authors.

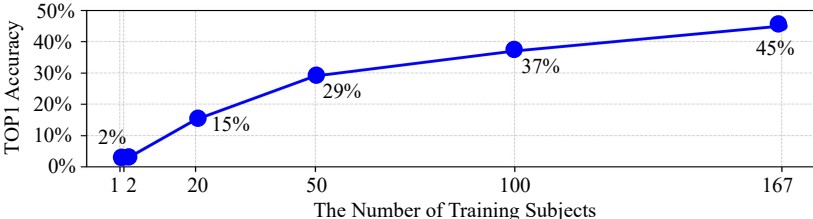

Figure 1: The performance on unseen subjects with the increase of the number of training subjects.

developing subject-specific models. Second, commonly used brain visual decoding datasets are built upon only a small number of participants. For example, the Natural Scenes Dataset (NSD) dataset (Allen et al., 2022) includes only 8 subjects. Most NSD-based studies (Kaur & Gandhi, 2019; Scotti et al., 2024a) employ only 4 of the 8 subjects, and use the NSDGeneral data, which contain only the manually mapped brain regions, rather than the entire brain data (See more detailed discussions in Sec. 3.1). Even those studies attempting to leverage multiple subjects are typically limited to less than 10 participants, and their networks are designed to handle only a small number of individuals. For instance, MindEye2 (Scotti et al., 2024b) and UMBRAE (Xia et al., 2024) use separate heads or tokenizers for different subjects. Therefore, the model becomes increasingly more complex with the increase of number of subject, which is hard to scale up to a larger number of subjects.

In this work, we aim to address this limitation and answer the question of whether brain decoding can be generalized to unseen subjects. (The term "generalization" refers to the model's ability to accurately decode information from unseen subjects, rather than merely performing inference.) To this end, we first consolidated an image-fMRI dataset, which consists of pairs of the stimulus image and the corresponding brain fMRI response. We build this dataset using the data from the Human Connectome Project (HCP) (Van Essen et al., 2013), which contains human brain neuroimages for various tasks. Among them, 177 subjects participated in the movie-viewing task, which provides the largest number of subjects available for extracting image-fMRI pairs for visual decoding study. In total, we collected 3,127 data pairs from 4 films watched by the 177 subjects. Compared to the commonly used datasets like NSD (8 subjects) (Allen et al., 2022) and BOLD5000 (4 subjects) (Chang et al., 2019), this dataset enables us to explore brain decoding performance with a much larger number of subjects. We consequently propose a new learning paradigm. Following MindEye1 (Scotti et al., 2024a), we use CLIP to encode the images, and employ a brain decoding network to map brain activities (characterized by fMRI voxels) into the same CLIP space by contrastive learning. To handle the varying fMRI voxel sizes across subjects, we normalize them to a common size through upsampling. Unlike previous methods that rely on specially designed input heads or subject-specific tokenizers, our paradigm uses the same processing for all subjects so that it can handle a large number of subjects without increasing the model complexity and parameters. (See the example in **Appendix**.)

We perform experiments on the fundamental retrieval task, which reflects well the capabilities of decoding models. Our method also has the potential to extend to reconstruction or grounding tasks with additional modules such as Stable Diffusion (Rombach et al., 2022). Through detailed experimentation, we uncover several important and intriguing findings. First, as shown in Fig. 1, the network demonstrates clear generalization ability as the number of training subjects increases, with top-1 accuracy rising from 2% (1 training subject) to 45% (167 training subjects) on unseen subjects (100 image-fMRI pairs). The accuracy can be further improved to 50% with additional training strategies. Second, the generalization capability holds for different network architectures. Using MLP, CNN and Transformer as the backbone, we achieve top-1 accuracies of 45%, 42%, and 34%, respectively, with 167 training subjects. Third, the generalization performance is influenced by subject similarity. We observe a bias when training on distinct groups such as gender, which represents one of the most easily identifiable categories of sample similarity. The model trained on 50 males achieves 36% top-1 accuracy on an unseen male subject, while the model trained on 50 females only obtains 27% top-1 accuracy on this test. Therefore, to explore further, we design an algorithm to calculate the similarity of fMRI responses among 167 individuals and train two models on the 20 most similar and 20 least similar subjects. The models achieve 21% and 2% top-1 accuracy, respectively, on an unseen subject, indicating the degree of similarity across subjects greatly affects the generalization performance. Our findings reveal that human brain activities share similarities, that require further exploration for generalization studies. It may be possible to train a large foundation model for brain decoding as bigger and more comprehensive datasets become available.

## 2 RELATED WORK

**Visual Brain Decoding.** With the advancement of deep learning (Radford et al., 2021; He et al., 2016; Vaswani et al., 2017; Rombach et al., 2022) and the emergence of high-quality fMRI datasets (Allen et al., 2022; Chang et al., 2019; Van Essen et al., 2013), many brain decoding methods with promising performance have been proposed. Takagi & Nishimoto (2023) utilized a latent diffusion model, specifically Stable Diffusion (Ho et al., 2020; Sohl-Dickstein et al., 2015), to reconstruct high-resolution images from fMRI data, preserving semantic fidelity without requiring additional training or fine-tuning. Brain-Diffuser (Ozcelik & VanRullen, 2023) improves the reconstruction process by first reconstructing basic image properties from fMRI signals and then refining the images using a latent diffusion model conditioned on multimodal features. Moreover, MindEye (Scotti et al., 2024a) encodes images using CLIP and then maps the corresponding fMRI data to the CLIP feature space, enabling strong image retrieval or reconstruction performance. However, most existing methods focus on decoding stimuli for individual subjects. While effective for individual decoding, they lack the generalization capability to new, unseen subjects.

**Visual Brain Decoding on Multiple Subjects.** Some brain decoding methods have been developed to leverage multiple subjects, which can be divided into two categories based on their objectives: (1) using multiple subjects to enhance subject-specific models, and (2) developing models that handle multiple subjects directly. For the first category, a straightforward way is to pre-train models on multiple subjects and then fine-tune them for individual subjects (Scotti et al., 2024b; Jiang et al., 2024; Qian et al., 2023; Ferrante et al., 2024). For example, MindEye2 (Scotti et al., 2024b) pre-trains the model on 7 subjects from the NSD dataset (Allen et al., 2022) and fine-tunes it on a different subject, using only 1/40 of the original data while achieving similar performance. The second category of methods aim to train a model with multiple subjects so that its performance on each subject (included in the training set) surpasses the models trained on each single subject. CLIP-MUSED (Zhou et al., 2024) and UMBRAE (Xia et al., 2024) are methods of this kind. However, most methods in this category still require separate heads or tokenizers for each subject. As the number of subjects increases, their training costs and model parameters grow linearly, making this approach impractical for larger subject pools. (We show an example in **Appendix**.) These multi-subject methods generally involve a limited number of subjects (less than 10), which are not sufficient enough for exploration. More importantly, while these methods demonstrate that certain information can be shared across subjects, they cannot generalize to unseen subjects.

**Subjects Alignment.** Some methods have been proposed to align new subjects to pre-trained models, known as subject alignment, to handle unseen subjects. Based on the alignment approach, these methods can be categorized into anatomical alignment (Jenkinson et al., 2002) and functional alignment (Haxby et al., 2011; Lorbert & Ramadge, 2012; Xu et al., 2012; Chen et al., 2015), among others. In visual brain decoding, the mainstream methods fall into functional alignment, which directly aligns the neural activity patterns across different subjects. For instance, Ferrante et al. (2024) used 1,000 common images viewed by 8 subjects from the NSD dataset to train an alignment model that maps other subjects to Subject 1. During inference, the brain signals of other subjects are converted into the format of Subject 1 and fed into the model trained on Subject 1. This approach can process new subjects with the model of existing subjects at a lower cost, yet it requires shared data for alignment. In this work, we aim to achieve model generalization without such alignment.

## 3 METHODS

In this section, we first describe how we consolidate the dataset for exploring generalizable visual brain decoding in Sec. 3.1. Then, we describe the proposed learning paradigm in Sec. 3.2. Finally, in Sec. 3.3 we outline how we calculate the subject similarity.

### 3.1 DATASET CONSOLIDATION

Most previous studies (Scotti et al., 2024a;b; Xia et al., 2024; Zhou et al., 2024) are conducted on datasets with fewer than 10 participants, which cannot be used to study whether visual decoding can be generalizable. Therefore, to explore the generalization capabilities of brain decoding models, the first step is to collect a dataset with a larger number of subjects. However, as shown in Tab. 1, current publicly available image-viewing datasets are limited in size. For example, the NSD dataset

Table 1: Summary of commonly used visual brain decoding datasets.

| Dataset | Task | Scanner | Subjects | Works Based on This Dataset |
|---|---|---|---|---|
| BOLD5000 (Chang et al., 2019) | image-viewing | 3T | 4 | Chen et al. (2023); Prince et al. (2022); Sexton & Love (2022) |
| GOD (Horikawa & Kamitani, 2017) | image-viewing | 3T | 5 | Chen et al. (2023); Du et al. (2023) |
| NSD (Allen et al., 2022) | image-viewing | 7T | 8 | MindEye1&2 (Scotti et al., 2024a;b); Gu et al. (2022); Ferrante et al. (2024); Qian et al. (2023); Han et al. (2024) |
| Raiders (Haxby et al., 2011) | movie-viewing | 3T | 21 | Chen et al. (2015); Shvartsman et al. (2018) |
| Forrest Gump (Hanke et al., 2014) | movie-viewing | 7T | 20 | Chen et al. (2015); Wagner et al. (2022); Huang et al. (2022) |
| Budapest (Matteo et al., 2020) | movie-viewing | 3T | 25 | Matteo et al. (2021); Busch et al. (2021) |
| HCP (Van Essen et al., 2013) | movie-viewing | 7T | 177 | Zhou et al. (2024); Lu et al. (2024) |

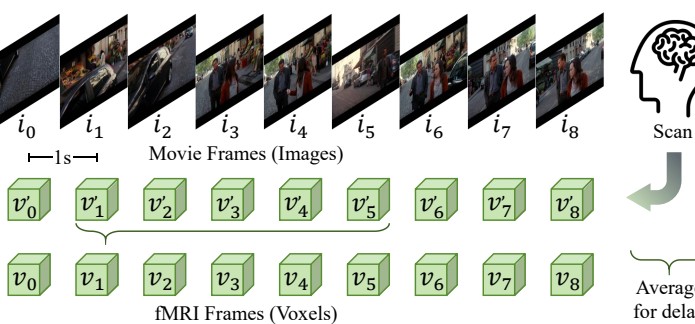

Figure 2: Dataset reconstruction from the HCP data. We extract the last frame $i$ in each second of the movie clip as the stimulus image, and average the fMRI voxels in the subsequent 4 seconds (due to hemodynamic delay) as the corresponding neural response $v$ to obtain image-fMRI pairs.

involves only 8 subjects (Allen et al., 2022) and BOLD5000 involves only 4 subjects (Chang et al., 2019). This is mainly due to the high costs, time demands, and challenges in keeping participants engaged during the long fMRI scanning sessions. Furthermore, these existing datasets are hard to be combined due to their significant differences in scanning equipment, resolution, design and post-processing methods. Even if combined, the total number of subjects remains very small.

We then turn to the movie-viewing task, which provides continuous, causally related visual inputs over a short period. Compared to the image-viewing task, movie-viewing can yield more data pairs in the same time-frame while keeping participants more engaged. Therefore, movie-viewing experiments can involve more subjects, as shown in Tab. 1. Actually, some studies have proposed to extract video frames for visual brain retrieval (Schneider et al., 2023) and classification (Zhou et al., 2024). Therefore, we propose to extract image-fMRI pairs from the movie-viewing task to build our dataset. Specifically, we choose the movie-viewing task in the HCP dataset, which involves 177 participants, making it the largest movie-viewing dataset available for visual decoding research. In data collection, the participants watched four audiovisual films, and the fMRI responses were captured with a repetition time (TR) of 1 second using a high-resolution 7T scanner. The volumetric images are registered to 1.6mm MNI space, with dimensions of $113 \times 136 \times 113$ per TR.

As shown in Fig. 2, to extract the corresponding image-fMRI pairs, we extract the last frame $i$ of each second of the film as the stimulus image, whose corresponding fMRI response voxel is denoted as $v'_i$. Following the 4-second hemodynamic delay suggested by Khosla et al. (2020), we average the fMRI signals from the subsequent 4 seconds to represent the neural response to each stimulus image (e.g., average $v'_1$ - $v'_5$ to obtain $v_1$ for $i_1$), for a total of 3,127 image-fMRI pairs for each subject. Finally, the reconstructed dataset includes 177 subjects with $177 \times 3{,}127$ image-fMRI pairs. By leveraging this dataset, we can explore the generalization capabilities of brain decoding models across a broader population. Our experiments in Sec. 4.2 demonstrate that generalization will emerge when a sufficient number of subjects are involved in training. This dataset provides a valuable foundation for researchers to investigate the behaviours of brain decoding models.

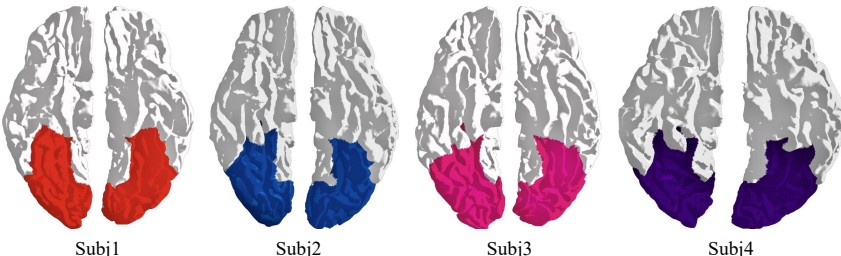

Subj1          Subj2          Subj3          Subj4

Figure 3: The visualization of scanned brain data in NSD dataset. The highlighted regions indicate the manually labeled NSDGeneral data. Compared to the whole brain, the NSDGeneral regions show significant variations across different subjects.

## 3.2 LEARNING PARADIGM

Prior studies are typically focused on decoding brain activity of individuals, while little work has been done on exploring the model generalization capability to unseen subjects. With our consolidated dataset in Sec. 3.1, we propose a learning paradigm to investigate the generalizability of visual brain decoding based on three core principles: (1) utilization of whole-brain data; (2) simple and flexible pipeline; (3) applicability to a large number of diverse subjects.

**Utilization of the Whole-Brain Data.**   As shown in Tab. 1, most recent visual brain decoding studies rely on the NSD dataset, which provides two types of training data: NSDGeneral data and whole-brain data. As illustrated in Fig. 3, the whole-brain data contain the fMRI voxels (about 800K elements) of the entire brain, while the NSDGeneral data comprise 1D vectors (flattened voxels) of only 10k–20k elements, which are manually labeled as vision-related brain regions, called NSDGeneral regions (see the highlighted areas). Since NSDGeneral data are directly related to brain regions in charge of visual processing, they often result in better visual decoding performance in the studies focused on single subjects. However, for research investigating the generalization capability across multiple subjects, we believe whole-brain data are more appropriate. First, the NSDGeneral data require manual segmentation, and hence they are difficult to scale across a large number of subjects, limiting their suitability for generalization studies. Note that most datasets, such as the HCP dataset, only provide whole-brain data. Second, as can be seen in Fig. 3, the manually labeled NSDGeneral regions show significant variations across different subjects, while the whole brain data (all the shown regions) show much less variation in shape. It is thus more difficult to train a common model for multiple subjects using the NSDGeneral data than the whole-brain data (See Sec. 4.5). Third, the NSDGeneral data exclude other brain regions, such as those in charge of memory, language, or contextual understanding. Ignoring those regions may prevent a more comprehensive decoding of brain activities (Zhou et al., 2024; Xu et al., 2021). Therefore, we advocate for using whole-brain data in the study of on model generalization, rather than data limited to specific brain regions.

**Simple and Flexible Pipeline.**   A simple and flexible learning pipeline is preferred to verify whether generalizability is a fundamental property of visual brain decoding, minimizing the factors brought by complex network designs. Our learning pipeline is shown in the left part of Fig. 4. The core idea is to project the paired stimulus-image $I$ and fMRI-voxel $V$ into the same feature space, where they could be as similar as possible. Following (Scotti et al., 2024a; Xia et al., 2024), we use the CLIP ViT-L/14 model to encode the images into features $F_I$, while the visual brain decoding network is trained to map fMRI-voxels to $F_V$ in the same feature space. The feature size of the CLIP embedding space is $257 \times 1024$, which retains detailed image information compared to the high-level semantic content of the final CLS token in CLIP. Contrastive learning is employed to align $F_I$ with $F_V$ using the CLIP Loss:

$$\mathcal{L} = \frac{1}{2N}\left(\sum_{i=1}^{N} -\log \frac{\exp(\text{sim}(F_I^i, F_V^i)/\tau)}{\sum_{j=1}^{N}\exp(\text{sim}(F_I^i, F_V^j)/\tau)} + \sum_{i=1}^{N} -\log \frac{\exp(\text{sim}(F_V^i, F_I^i)/\tau)}{\sum_{j=1}^{N}\exp(\text{sim}(F_V^i, F_I^j)/\tau)}\right),$$
$$(1)$$

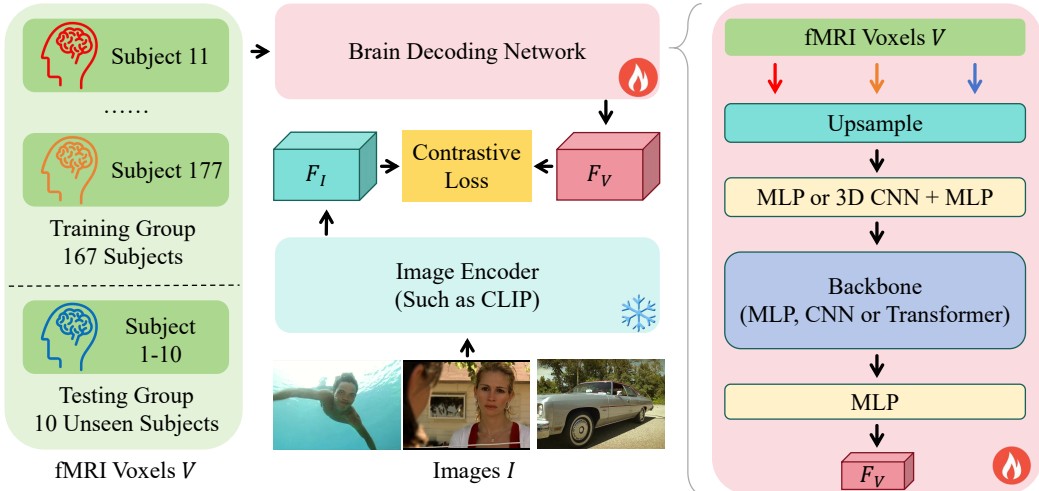

Figure 4: The overview of our learning pipeline (left) and visual brain decoding network (right).

where $F_I^i$ and $F_V^i$ are the embeddings of the $i^{th}$ image and fMRI voxel, $\tau$ is the temperature parameter, and $\text{sim}(x, y)$ represents the cosine similarity between $x$ and $y$:

$$\text{sim}(x, y) = \frac{x \cdot y}{\|x\|\|y\|}. \tag{2}$$

During inference, retrieval is performed by calculating the cosine similarity and taking the most similar pairs. We use MLP as the decoding network in most of our experiments, while the network architecture can be changed to CNN and Transformer (See Sec. 4.3), and the model performance can be further improved with additional strategies (See **Appendix**).

**Applicability to Many Diverse Subjects.** Previous studies are mostly focused on a small number of participants, and they use separate heads or tokenizers for different subjects to improve performance. While being effective in small-scale studies, these approaches become impractical and cannot scale up as the number of subjects increases. To explore model generalizability, we employ the same decoding network (see the right part of Fig. 4) to accommodate a large number of subjects without requiring specific adaptations for each individual. Due to the structural differences in brain anatomy, the size of fMRI voxels, even for the same brain activity, can vary across subjects, which cannot be directly batched for network training. To solve this issue, we apply simple upsampling to resize the voxels to a standardized larger size. This method is simple and straightforward and can be done in the processing of the dataset. Experimental results show that this unified approach does not compromise performance in whole-brain decoding and can even enhance performance across multiple subjects (See Sec. 4.5). As shown in the right of Fig. 4, our final network design involves an upsample layer to normalize voxel sizes for all subjects, followed by feeding the data into a unified network without requiring any subject-specific adaptations.

## 3.3 GENERALIZATION PERFORMANCE VS. SUBJECT SIMILARITY

During experiments, we notice some performance biases of models trained on different gender groups, which represent one of the most easily identifiable similarity categories. It inspires us to hypothesize that the degree of similarity among subjects might impact the model generalization performance. To test this hypothesis, we need to identify which subjects are more similar to the given subjects. For a target subject $S_t$, given a set of images $I$ viewed by $N$ different subjects $S_N$, for each image $i$, we can calculate the cosine similarity (refer to Eq. 2) between the fMRI voxel $v_{i,S_t}$ of target subject and the voxel $v_{i,S_n}$ of subject $S_n \in S_N$ as follow: $Sim\_score_{i,S_t,S_n} = sim(v_{i,S_t}, v_{i,S_n})$. The similarity score reflects the likeness between the two subjects ($S_t$ and $S_n$) based on a given image $i$. The overall similarity score of the two subjects can be obtained by averaging over all images $I$. However, the outlier images can make the averaged score less robust. Therefore, we use a rank-based method. For each image $i$, we calculate $Sim\_score_{i,S_t,S_n}$ and rank the $N$ scores from highest to lowest. Then, we select the top 10 subjects based on their ranks and award them one

Table 2: Results of models trained on our consolidated HCP dataset with different number of training subjects. TOP1 Acc. and TOP3 Acc. are averaged over the unseen subjects (Subjs 1-10). TOP1 Acc. (seen) is averaged over all seen subjects participated in training.

| Training Subjects | No. of Training Subjects | TOP1 Acc. | TOP3 Acc. | TOP1 Acc. (seen) |
|---|---|---|---|---|
| Subj 11 | 1 | 2% | 5% | 79% |
| Subjs 11-12 | 2 | 2% | 6% | 84% |
| Subjs 11-30 | 20 | 15% | 29% | 83% |
| Subjs 11-60 | 50 | 29% | 43% | 83% |
| Subjs 11-110 | 100 | 37% | 52% | 82% |
| Subjs 11-177 | 167 | 45% | 61% | 82% |

rank credit. After repeating this process for all images, subjects with higher total rank credits are considered more similar to the target subject $S_t$. The process to calculate the rank credit of $S_n$ for $S_t$ can be formulated as:

$$\text{Rank\_Credit}(S_t, S_n) = \sum_{i=1}^{I} \mathbb{1}(S_n \in \text{top\_10\_rank}(Sim\_score_{i,S_t,S_j} \text{ for } j = 1, 2 \ldots, N)), \quad (3)$$

where $\mathbb{1}(\cdot)$ is an indication function that assigns 1 if $S_n$ is among the top 10 most similar subjects to $S_t$ based on image $i$, otherwise 0. Finally, we use models trained on both similar and dissimilar subjects to explore how subject similarity influences the generalization performance. The results are shown in Sec. 4.4.

## 4 EXPERIMENTS

### 4.1 IMPLEMENTATION DETAILS

We implement all models using PyTorch (Paszke et al., 2017). Except specifically indicated, we employ MLP and 3D CNN as the backbone for feature extraction when using whole-brain data. The detailed network structure can be found in **Appendix**. During training, we employ the CLIP loss (Radford et al., 2021) and the AdamW optimizer (Loshchilov & Hutter, 2017) to optimize the models ($\beta_1 = 0.9$, $\beta_2 = 0.999$). We set the batch size to 300, and apply the OneCycleLR strategy with a warm-up phase to adjust the learning rate, with a maximum learning rate of $1 \times 10^{-4}$. The HCP dataset we consolidated includes 177 subjects, each subject having 3,127 image-fMRI pairs. We randomly choose 100 images and the corresponding fMRI voxels as the test pairs, and use the rest as the training pairs. Note that the test pairs of all subjects are from the same 100 images. Subjs 1-10 are designated as unseen subjects, with the remaining 167 subjects as seen subjects.

In our experiments, several models will be trained on different numbers of subjects. For convenience of expression, we define one training epoch based on the number of image-fMRI pairs of a single subject; that is, one epoch contains 3,027 image-fMRI pairs. The numbers of epochs to train models on 1, 2, 20, 50, 100, and 167 seen subjects are 200, 200, 400, 600, 800, and 1,000, respectively. For the experiment on the NSD dataset, which includes 8 subjects, we follow the standard train/test split with 1,000 test images (Allen et al., 2022), and select Subj 2 and Subj 5 as unseen subjects. We train the models with 1 and 6 seen subjects for 120 and 360 epochs, respectively, and each epoch includes 9,000 image-fMRI pairs.

### 4.2 MAIN RESULTS ON GENERALIZATION PERFORMANCE

As describe in Sec. 4.1, we train models on 1, 2, 20, 50, 100 and 167 subjects and evaluate them on 10 unseen subjects (Subjs 1-10). The results are shown in Tab. 2. We can clearly see that as the number of training subjects increases, the model's generalization capability on unseen subjects improves. When only one or two subjects are used in training, the generalization capability is weak. When the number of training subjects reaches 167, the TOP1 and TOP3 accuracies improve to 45% and 61%, respectively. (TOP1 and TOP3 accuracies indicate the probability that the correct sample is included among the top 1 or top 3 most relevant retrieval results returned by the network.)

| Backbones | TOP1 Acc. | TOP3 Acc. | TOP1 Acc. (seen) |
|-----------|-----------|-----------|------------------|
| MLP | 45% | 61% | 82% |
| 1D CNN | 42% | 58% | 80% |
| 3D CNN | 40% | 57% | 80% |
| Transformer | 34% | 52% | 77% |

Table 3: Results of models with different backbones trained on 167 subjects. TOP1 Acc. and TOP3 Acc. are averaged over ten unseen subjects (Subjs 1-10). TOP1 Acc. (seen) is averaged over all seen subjects.

Considering that our test set contains 100 image-fMRI pairs, such a generalization performance is highly encouraging. Fig. 1 plots the curve of TOP1 accuracy vs. the number of training subjects. We observe that the generalization performance continues to improve steadily. Even with 167 subjects, it does not reach a plateau. This suggests that the models hold potential for further improvement if more subjects can be introduced for training.

We also provide in Tab. 2 the results of our model on seen subjects for reference. The test pairs are from the subjects involved in the training process. We can see that the TOP1 accuracies are very close for different number of training subjects. This is reasonable because the testing data share similar distribution with the training data, regardless of whether 1 or 167 subjects are involved, and the network is able to fit the distribution via sufficient training.

### 4.3 THE GENERALIZATION PERFORMANCE WITH DIFFERENT BACKBONES

In Sec. 4.2, we used MLP as the backbone and validated the generalization capability of our models trained on more subjects. Here, we validate whether this conclusion holds for other popular network architectures, such as CNN and Transformer networks. The details of the employed network architectures can be found in **Appendix**. The results are shown in Tab. 3, which demonstrates that while there are some differences in performance, the generalization capability is consistently achieved across different network architectures. Specifically, the MLP achieves the best generalization performance, with 45% TOP1 accuracy on unseen subjects and 82% on seen subjects, followed closely by 1D CNN and 3D CNN, which yield comparable results. The Transformer network exhibits the lowest performance, which may be attributed to the fact that Transformer typically needs larger training datasets to exhibit its superiority, whereas our current dataset for visual brain decoding is relatively small, making CNNs and MLPs more effective in this case.

### 4.4 GENERALIZATION VS. SUBJECT SIMILARITY

From the experiments in previous sections, we have seen that the network could exhibit obvious generalization capability when enough subjects are used in training. During our experiments, interestingly, we notice some performance biases of models trained on different gender groups. Gender is one of the most commonly observed characteristics of sample similarity, suggesting that the similarity among subjects might impact the model generalization performance. To be specific, we train three models using data from 50 male subjects, 50 female subjects, and a mixed group of 25 male and 25 female subjects, respectively. Then, we evaluate these models on unseen male Subj 1 and female Subj 2, and the results are shown in Tab. 4. One can see that the model trained on male subjects achieves the best retrieval performance on the unseen male subject (Subj 1) with 36% TOP1 accuracy and 60% TOP3 accuracy, but it performs the worst on the unseen female subject (Subj 2) with 25% TOP1 accuracy and 37% TOP3 accuracy. In contrast, the model trained on female subjects shows the opposite behaviour, performing better on Subj 2 and worse on Subj 1. Meanwhile, the model trained on the mixed group always obtain the intermediate result on the unseen subjects. Such results suggest that the generalization capability is related to the similarity among subjects.

Therefore, we further explore this phenomenon by finding similar and dissimilar subjects to Subj 1 and Subj 2. Utilizing the method described in Sec. 3.3, we identify 20 most similar subjects and 20 least similar subjects to Subj 1, as well as 20 most similar and least similar subjects to Subj 2. As shown in Tab. 4, the model trained on the 20 subjects most similar to Subj 1 achieves the best performance on Subj 1, with a TOP1 accuracy of 21% and a TOP3 accuracy of 36%. In contrast, the model trained on the 20 most dissimilar subjects performs significantly worse, with a TOP1 accuracy of 8% and a TOP3 accuracy of 14%. A similar trend can be observed for Subj 2, where the model trained on the 20 most similar / dissimilar subjects achieves the highest / lowest

Table 4: Results of models trained on subjects that have different gender or similarity. All the training subjects are from Subj 3-167. Best and worst results are marked by red and blue.

| Training Subjects | TOP1 Acc. on Subj 1 (male) | TOP3 Acc. on Subj 1 (male) | TOP1 Acc. on Subj 2 (female) | TOP3 Acc. on Subj 2 (female) |
|---|---|---|---|---|
| 50 male | 36% | 60% | 25% | 37% |
| 50 female | 27% | 40% | 30% | 42% |
| 25 male + 25 female | 32% | 49% | 27% | 40% |
| 20 similar to Subj 1 | 21% | 36% | - | - |
| 20 dissimilar to Subj 1 | 8% | 14% | - | - |
| 20 similar to Subj 2 | - | - | 21% | 31% |
| 20 dissimilar to Subj 2 | - | - | 2% | 7% |
| Subjs 11-30 (20) | 17% | 31% | 18% | 23% |
| 20 similar + 20 dissimilar to Subj 1 | 21% | 30% | - | - |
| 20 similar + 20 dissimilar to Subj 2 | - | - | 20% | 28% |

Table 5: Results of models trained on NSD dataset. As in previous works, we randomly extracted 300 pairs from 1,000 pairs of the test set to perform the testing, and run 30 times the experiments to take the average. 'MindEye1' means our implemented model of MindEye1 (Scotti et al., 2024a).

| Models | Training Subjects | Testing Subjects | NSDGeneral Data | | NSD Whole-brain Data | |
|---|---|---|---|---|---|---|
| | | | Data Format | TOP1 Acc. | Data Format | TOP1 Acc. |
| MindEye1 | Subj 1 | Subj 1 | $1 \times 15{,}724$ | 85% | $83 \times 104 \times 81$ | 35% |
| Ours | Subj 1 | Subj 1 | $1 \times 18{,}000$ | 86% | $113 \times 136 \times 113$ | 46% |
| MindEye1 | Subj 7 | Subj 7 | $1 \times 12{,}682$ | 70% | $81 \times 95 \times 78$ | 23% |
| Ours | Subj 7 | Subj 7 | $1 \times 18{,}000$ | 70% | $113 \times 136 \times 113$ | 29% |
| Ours | Subjs 1,3,4,6,7,8 | Subj 1 | $1 \times 18{,}000$ | 83% | $113 \times 136 \times 113$ | 49% |
| Ours | Subjs 1,3,4,6,7,8 | Subj 7 | $1 \times 18{,}000$ | 69% | $113 \times 136 \times 113$ | 35% |
| Ours | Subjs 1,3,4,6,7,8 | Subjs 2,5 | $1 \times 18{,}000$ | 1% | $113 \times 136 \times 113$ | 1% |

performance. Additionally, the model trained on Subjs 11-30, as a reference for randomly selected 20 subjects, yields moderate performance on both Subj 1 and Subj 2. This demonstrates that the similarity between subjects can largely affect the generalization performance. Even with 20 subjects in training, if the subjects are not similar, the model can achieve little generalization capability, such as 2% TOP1 Acc. on Subj 2. We also train models on a mixed set of 20 similar and 20 dissimilar subjects for Subj 1 and Subj 2. The results closely match the performance of models trained on the 20 similar subjects alone.

The above experimental results show that when the subjects are similar, the models achieve better generalization performance, and vice versa. On the other hand, when a mix of similar and dissimilar subjects are used for training, generalization remains stable, with performance approaching to the models trained on similar subjects. This suggests that generalization capability depends on learning inherent commonalities among human brains, with substantial tolerance for dissimilarities. It also explains why increasing the number of subjects enhances generalization — a larger dataset is more likely to include subjects with higher similarities.

## 4.5 EXPERIMENTAL RESULTS ON THE NSD DATASET

To demonstrate the flexibility of our pipeline, we also train the model on the NSD dataset, including both NSDGeneral and whole-brain data. The results are shown in Tab. 5. We see that the models trained on Subj 1 using NSDGeneral data with both the original data format (*i.e.*, $1 \times 15{,}724$ in MindEye1) and our normalized data format (*i.e.*, $1 \times 18{,}000$) achieve similar TOP1 accuracy, *i.e.*, 85% and 86%, respectively. (The TOP1 accuracy reported in the original paper of MindEye1 (Scotti et al., 2024a) is 84%.) The results on Subj 7 can yield similar conclusion. This demonstrates that our

pipeline can be well applied to the individual-specific scenario using NSDGeneral data. However, by using simple interpolation based upsampling to normalize the data format as a $1 \times 18,000$ vector, we can train models on multiple subjects with different original NSDGeneral data sizes. As shown in the bottom three rows of Tab. 5, by training on subjs 1,3,4,6,7,8 and testing on subj 1 or subj 7, 83% and 69% TOP1 accuracy can still be obtained since subj 1 or subj 7 are included in the training data. However, when testing on the unseen subjs 2 and 5, only 1% TOP1 accuracy is obtained. This is expected, as the NSD dataset consists of only 8 subjects, which is too limited to ensure robust model generalization performance.

Finally, we evaluate the models trained with whole-brain data. The top right panel of Tab. 5 shows the results on Subj 1 and Subj 7 by MindEye1, which uses the original whole-brain data format, and our model, which uses the normalized data format (*i.e.*, $113 \times 136 \times 113$). We see that in the case of whole-brain data, the simple normalization of data size can improve much the TOP1 accuracy from 35% to 46% for subj 1 and from 23% to 29% for subj 7. The accuracy is lower than that on NSDGeneral data because the NSDGeneral data are manually labeled brain visual regions. Again, the normalized data size enables us to train models on multiple subjects with different original data sizes, which cannot be done by MindEye1. As shown in the bottom right panel of Tab. 5, our model trained on six subjects achieves 49% TOP1 accuracy on Subj 1 and 35% on Subj 7, outperforming single-subject models trained on Subj 1 (46%) and Subj 7 (29%), respectively. In contrast, on NSDGeneral data, the multiple-subject models perform slightly worse than their single-subject counterparts. This discrepancy highlights the individual-specific nature of NSDGeneral data, as discussed in Sec. 3.1 (see also Fig. 3). We also report the generalization performance of models trained on 6 subjects on unseen Subjs 2 and 5. Again, the model shows weak generalization ability due to the small number of training subjects in NSD dataset.

## 5 DISCUSSION: THE SOURCE OF GENERALIZATION ABILITY

Based on the experimental results, we observe that when a sufficient number of subjects are included during training, various networks can exhibit generalization capabilities on unseen subjects. In this section, we discuss the potential causes underlying this generalization ability. Firstly, we assert that generalization does not arise from a sophisticated upsampling (alignment) module or any other similar mechanism. Instead, we argue that it originates from "involving" rather than merely "aligning" the patterns across different subjects. In other words, the reason why our method performs well on unseen subjects is that the training data likely encompass the feature-mapping patterns of unseen subjects. This hypothesis is supported by our following experimental findings. (1) Increasing Training Subjects in in Sec. 4.2: As the number of training subjects increases, generalization improves. It is likely because that the training set tends to include representative patterns of unseen subjects. (2) Subject Similarity in Sec. 4.4: When the number of training subjects is fixed, training groups that share greater similarity with the testing subjects (*e.g.*, in terms of gender) tend to result in better generalization performance. Our main contribution is to highlight this novel perspective: generalization ability can be obtained from "involving" a sufficiently diverse set of subjects, rather than relying on alignment or fine-tuning mechanisms; and generalization also depends on subject similarities.

## 6 CONCLUSION

Previous visual brain decoding studies typically focus on individual subjects, or training with multiple-subjects but decoding on seen subjects, while little work has been done to explore the possibility of generalizing visual brain decoding on unseen subjects. We attempt to fill this gap by leveraging a large dataset from the Human Connectome Project (HCP), constructing $177 \times 3,127$ image-fMRI pairs from 177 subjects. Using this dataset, we proposed a learning paradigm, which utilized whole-brain data and a simple and uniform pipeline for processing all subjects, without requiring individual-specific adaptations. Via detailed experiments, we found that the model generalization capability emerged with the increase of training subjects, and such generalization capability hold across different network architectures. In addition, the similarity between subjects also played a role in improving the generalization capability. These findings reveal the inherent similarity in brain activities across individuals, which has significant implications for future studies. As larger and more diverse datasets may become available, this work can provide a basis for training a brain encoding foundation model for the future.

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
