# Toward Generalizing Visual Brain Decoding to Unseen Subjects: Appendix

**Xiangtao Kong** [1,*] **Kexin Huang** [1,*], **Ping Li** [1,†] **Lei Zhang** [1,†]
[1] The Hong Kong Polytechnic University
{xiangtao.kong, kexin0.huang}@connect.polyu.hk,
ping2.li@polyu.edu.hk, cslzhang@comp.polyu.edu.hk

In this Appendix, we first provide the details of the networks used in the main paper. Second, we outline the details of the BiMixCo+SoftCLIP training strategy used in Sec. 4.5 of the main paper. Finally, we present the visualization of the image retrieval results.

## 1 The Details of Architectures

In the main paper, we validated the generalization capability of visual brain decoding models on unseen subjects across popular network architectures, including MLP, CNN and Transformer. We show the detailed network structures in this section. The input to the networks is fMRI voxels of size $113 \times 136 \times 113$.

Table 1: Detailed architecture of our MLP network.

| Layer Type | Input Size | Output Size | Kernel/Linear Size | Stride/Padding |
|---|---|---|---|---|
| Conv3D (conv1) | $1 \times 113 \times 136 \times 113$ | $32 \times 38 \times 46 \times 38$ | $9 \times 9 \times 9$ | Stride 3, Padding 4 |
| BatchNorm3D (bn1) | $32 \times 38 \times 46 \times 38$ | $32 \times 38 \times 46 \times 38$ | - | - |
| Conv3D (conv2) | $32 \times 38 \times 46 \times 38$ | $48 \times 19 \times 23 \times 19$ | $7 \times 7 \times 7$ | Stride 2, Padding 3 |
| BatchNorm3D (bn2) | $48 \times 19 \times 23 \times 19$ | $48 \times 19 \times 23 \times 19$ | - | - |
| Conv3D (conv3) | $48 \times 19 \times 23 \times 19$ | $64 \times 10 \times 12 \times 10$ | $5 \times 5 \times 5$ | Stride 2, Padding 2 |
| BatchNorm3D (bn3) | $64 \times 10 \times 12 \times 10$ | $64 \times 10 \times 12 \times 10$ | - | - |
| ReLU (activation) | $64 \times 10 \times 12 \times 10$ | $64 \times 10 \times 12 \times 10$ | - | - |
| Flatten | $64 \times 10 \times 12 \times 10$ | 76800 | - | - |
| Linear (lin0) | 76800 | 4096 | - | - |
| MLP (4 layers) | 4096 | 4096 | - | - |
| Linear (lin1) | 4096 | $257 \times 1024$ | - | - |

**MLP.** Tab. 1 presents the architecture of our MLP network. We first employ a 3-layer 3D CNN for feature extraction, followed by a fully connected layer to reduces the feature to a 1D vector of 4096 elements. Similar to MindEye1 (Scotti et al., 2024a), we apply the MLP backbone in this feature space with the vector length of 4096. Finally, we utilize a fully connected layer to expand the feature from 4096 to $257 \times 1024$, which is used for loss calculation.

**1D CNN.** As shown in Tab. 2, similar to our MLP network, our 1D CNN network first reduces the feature to a vector of length 4096. Then we use several separate 1D CNN layers and eight 1D ResBlock layers for feature mapping, which constitute our 1D CNN backbone. Finally, a fully connected layer is used to produce the output with size of $257 \times 1024$.

**3D CNN.** Different from the above MLP and 1D CNN networks, as shown in Tab. 3, our 3D CNN network uses 3D CNN layers and 3D ResBlock layers to process the input. After the main processing, a fully connected layer is used to produce the final output with size of $257 \times 1024$.

---

[*] Equal contributions.
[†] Corresponding authors.

Table 2: Detailed architecture of our 1D CNN network.

| Layer Type | Input Size | Output Size | Kernel/Linear Size | Stride/Padding |
|---|---|---|---|---|
| Conv3D (conv1) | $1 \times 113 \times 136 \times 113$ | $32 \times 38 \times 46 \times 38$ | $9 \times 9 \times 9$ | Stride 3, Padding 4 |
| BatchNorm3D (bn1) | $32 \times 38 \times 46 \times 38$ | $32 \times 38 \times 46 \times 38$ | - | - |
| Conv3D (conv2) | $32 \times 38 \times 46 \times 38$ | $48 \times 19 \times 23 \times 19$ | $7 \times 7 \times 7$ | Stride 2, Padding 3 |
| BatchNorm3D (bn2) | $48 \times 19 \times 23 \times 19$ | $48 \times 19 \times 23 \times 19$ | - | - |
| Conv3D (conv3) | $48 \times 19 \times 23 \times 19$ | $64 \times 10 \times 12 \times 10$ | $5 \times 5 \times 5$ | Stride 2, Padding 2 |
| BatchNorm3D (bn3) | $64 \times 10 \times 12 \times 10$ | $64 \times 10 \times 12 \times 10$ | - | - |
| ReLU (activation) | $64 \times 10 \times 12 \times 10$ | $64 \times 10 \times 12 \times 10$ | - | - |
| Flatten | $64 \times 10 \times 12 \times 10$ | 76800 | - | - |
| Linear (lin0) | 76800 | 4096 | - | - |
| Reshape | 4096 | $1 \times 4096$ | - | - |
| Conv1D (conv01) | $1 \times 4096$ | $64 \times 680$ | Kernel 13 | Stride 6 |
| BatchNorm1D (bn01) | $64 \times 680$ | $64 \times 680$ | - | - |
| Conv1D (conv02) | $64 \times 680$ | $128 \times 134$ | Kernel 11 | Stride 5 |
| BatchNorm1D (bn02) | $128 \times 134$ | $128 \times 134$ | - | - |
| Conv1D (conv03) | $128 \times 134$ | $256 \times 32$ | Kernel 9 | Stride 4 |
| BatchNorm1D (bn03) | $256 \times 32$ | $256 \times 32$ | - | - |
| Conv1D (conv04) | $256 \times 32$ | $512 \times 9$ | Kernel 7 | Stride 3 |
| BatchNorm1D (bn04) | $512 \times 9$ | $512 \times 9$ | - | - |
| ResBlock1D (x8) | $512 \times 9$ | $512 \times 9$ | - | - |
| Flatten | $512 \times 9$ | 4608 | - | - |
| Linear (lin1) | 4608 | $257 \times 1024$ | - | - |

Table 3: Detailed architecture of our 3D CNN network.

| Layer Type | Input Size | Output Size | Kernel/Linear Size | Stride/Padding |
|---|---|---|---|---|
| Conv3D (conv1) | $1 \times 113 \times 136 \times 113$ | $32 \times 38 \times 46 \times 38$ | $9 \times 9 \times 9$ | Stride 3, Padding 4 |
| BatchNorm3D (bn1) | $32 \times 38 \times 46 \times 38$ | $32 \times 38 \times 46 \times 38$ | - | - |
| Conv3D (conv2) | $32 \times 38 \times 46 \times 38$ | $48 \times 19 \times 23 \times 19$ | $7 \times 7 \times 7$ | Stride 2, Padding 3 |
| BatchNorm3D (bn2) | $48 \times 19 \times 23 \times 19$ | $48 \times 19 \times 23 \times 19$ | - | - |
| Conv3D (conv3) | $48 \times 19 \times 23 \times 19$ | $64 \times 10 \times 12 \times 10$ | $5 \times 5 \times 5$ | Stride 2, Padding 2 |
| BatchNorm3D (bn3) | $64 \times 10 \times 12 \times 10$ | $64 \times 10 \times 12 \times 10$ | - | - |
| Conv3D (conv4) | $64 \times 10 \times 12 \times 10$ | $90 \times 5 \times 6 \times 5$ | $5 \times 5 \times 5$ | Stride 2, Padding 2 |
| BatchNorm3D (bn4) | $90 \times 5 \times 6 \times 5$ | $90 \times 5 \times 6 \times 5$ | - | - |
| Conv3D (conv5) | $90 \times 5 \times 6 \times 5$ | $150 \times 3 \times 3 \times 3$ | $5 \times 5 \times 5$ | Stride 2, Padding 2 |
| BatchNorm3D (bn5) | $150 \times 3 \times 3 \times 3$ | $150 \times 3 \times 3 \times 3$ | - | - |
| ReLU (activation) | $150 \times 3 \times 3 \times 3$ | $150 \times 3 \times 3 \times 3$ | - | - |
| ResBlock3D (x8) | $150 \times 3 \times 3 \times 3$ | $150 \times 3 \times 3 \times 3$ | - | - |
| Flatten | $150 \times 3 \times 3 \times 3$ | 4050 | - | - |
| Linear (lin1) | 4050 | $257 \times 1024$ | - | - |

**Transformer.** The details of our Transformer network are shown in Tab. 4. Similar to our MLP and 1D CNN networks, our Transformer network first reduces the feature to a 4096 sized vector. Then we reshape the feature to $16 \times 256$, which is then sent to the Transformer backbone with 24 Transformer block layers. After that, we reshape the output of Transformer backbone back to 4096 and employ a fully connected layer to produce the final output with size of $257 \times 1024$.

## 2 TRAINING STRATEGY

**Results.** In our main experiments, in order to prove that the generalization capability does not come from some specific strategies, we use the simplest contrastive learning pipeline and the CLIP loss to verify our approach on commonly used network architectures. In this section, we demonstrate that the model performance can be further enhanced if stronger training strategies can be employed. In particular, we adopt the BiMixCo+SoftCLIP training strategy (using BiMixCo for one-third of the training and employs SoftCLIP for the rest of training) from MindEye1 (Scotti et al., 2024a).

Table 4: Detailed architecture of our Transformer network.

| Layer Type | Input Size | Output Size | Kernel/Linear Size | Stride/Padding |
|---|---|---|---|---|
| Conv3D (conv1) | $1 \times 113 \times 136 \times 113$ | $32 \times 38 \times 46 \times 38$ | $9 \times 9 \times 9$ | Stride 3, Padding 4 |
| BatchNorm3D (bn1) | $32 \times 38 \times 46 \times 38$ | $32 \times 38 \times 46 \times 38$ | - | - |
| Conv3D (conv2) | $32 \times 38 \times 46 \times 38$ | $48 \times 19 \times 23 \times 19$ | $7 \times 7 \times 7$ | Stride 2, Padding 3 |
| BatchNorm3D (bn2) | $48 \times 19 \times 23 \times 19$ | $48 \times 19 \times 23 \times 19$ | - | - |
| Conv3D (conv3) | $48 \times 19 \times 23 \times 19$ | $64 \times 10 \times 12 \times 10$ | $5 \times 5 \times 5$ | Stride 2, Padding 2 |
| BatchNorm3D (bn3) | $64 \times 10 \times 12 \times 10$ | $64 \times 10 \times 12 \times 10$ | - | - |
| ReLU (activation) | $64 \times 10 \times 12 \times 10$ | $64 \times 10 \times 12 \times 10$ | - | - |
| Flatten | $64 \times 10 \times 12 \times 10$ | 76800 | - | - |
| Linear (lin0) | 76800 | 4096 | - | - |
| Reshape | 4096 | $16 \times 256$ | - | - |
| Transformer (d=256, h=8) (x24) | $16 \times 256$ | $16 \times 256$ | - | - |
| Reshape | $16 \times 256$ | 4096 | - | - |
| Linear (lin1) | 4096 | $257 \times 1024$ | - | - |

Table 5: Results of models trained with different training strategies. TOP1 Acc. is averaged over unseen subjects (Subjs 1-10). TOP1 Acc. (seen) is averaged over all seen subjects during training.

| Trainging Subjects | Subject Number | Strategies | TOP1 Acc. | TOP1 Acc. (seen) |
|---|---|---|---|---|
| Subj 11 | 1 | CLIP / BiMixCo+SoftCLIP | 2% / 6% | 79% / 83% |
| Subjs 11-177 | 167 | CLIP / BiMixCo+SoftCLIP | 45% / 50% | 82% / 84% |

The BiMixCo+SoftCLIP strategy incorporates a data augmentation technique, extending the mixup approach with the InfoNCE loss (He et al., 2020). It also replaces the CLIP loss with the SoftCLIP loss, which leverages softmax probability distributions rather than hard labels.

As shown in Tab. 5, the adoption of BiMixCo+SoftCLIP leads to a noticeable improvement in generalization performance for both single-subject models (TOP1 accuracy improves from 2% to 6%) and multiple-subject models (TOP1 accuracy improves from 45% to 50%). This strategy also enhances the retrieval accuracy on seen subjects. These results suggest that better learning strategies can be designed to boost the model generalization capabilities, highlighting the potential of our approach for visual brain decoding. Details of these methods can be found below:

**BiMixCo.** The BiMixCo combines MixCo (Kim et al., 2020) (an extension of mixup that uses the InfoNCE loss) and the bidirectional CLIP loss. To be specific, as described in MindEye1 (Scotti et al., 2024a), voxels are mixed by a factor $\lambda$ sampled from the Beta distribution with $\alpha = \beta = 0.15$:

$$x_{\text{mix}_{i,k_i}} = \lambda_i \cdot x_i + (1 - \lambda_i) \cdot x_{k_i}, \quad p_i^* = f(x_{\text{mix}_{i,k_i}}), \quad p_i = f(x_i), \quad t_i = \text{CLIP}_{\text{Image}}(y_i), \quad (1)$$

where $x_i$ and $y_i$ represent the $i$-th fMRI voxel and image, respectively. $k_i \in [1, N]$ is an arbitrary mixing index for the $i$-th datapoint and $f$ represents the decoding network. $p^*$, $p$ and $t$ are $L_2$-normalized. The CLIP loss with MixCo is defined as:

$$\mathcal{L}_{\text{BiMixCo}} = -\sum_{i=1}^{N} \left[ \lambda_i \cdot \log \left( \frac{\exp \left( \frac{p_i^* \cdot t_i}{\tau} \right)}{\sum_{m=1}^{N} \exp \left( \frac{p_i^* \cdot t_m}{\tau} \right)} \right) + (1 - \lambda_i) \cdot \log \left( \frac{\exp \left( \frac{p_i^* \cdot t_{k_i}}{\tau} \right)}{\sum_{m=1}^{N} \exp \left( \frac{p_i^* \cdot t_m}{\tau} \right)} \right) \right]$$
$$-\sum_{j=1}^{N} \left[ \lambda_j \cdot \log \left( \frac{\exp \left( \frac{p_j^* \cdot t_j}{\tau} \right)}{\sum_{m=1}^{N} \exp \left( \frac{p_m^* \cdot t_j}{\tau} \right)} \right) + \sum_{\{l | k_l = j\}} (1 - \lambda_l) \cdot \log \left( \frac{\exp \left( \frac{p_l^* \cdot t_j}{\tau} \right)}{\sum_{m=1}^{N} \exp \left( \frac{p_m^* \cdot t_j}{\tau} \right)} \right) \right],$$
$$(2)$$

where $\tau$ is a temperature hyperparameter, and $N$ is the batch size.

**SoftCLIP.** The soft contrastive loss (Scotti et al., 2024a) takes the dot product of CLIP image embeddings within a batch to generate the soft labels. The loss (we omit the bidirectional component

Table 6: Model parameters of UMBRAE (Xia et al., 2024) and our method with different subject number. Both the methods use the same backbones of UMBRAE (Xia et al., 2024).

| Number of Testing Subjects | UMBRAE (Xia et al., 2024) | Ours |
|:---:|:---:|:---:|
| 1 | 112.63M | 112.63M |
| 10 | 213.43M | 112.63M |
| 100 | 1221.42M | 112.63M |
| 1000 | 11310.43M | 112.63M |

for brevity) is calculated between CLIP-CLIP and Brain-CLIP matrices as:

$$\mathcal{L}_{\text{SoftCLIP}} = -\sum_{i=1}^{N}\sum_{j=1}^{N}\left[\frac{\exp\left(\frac{t_i \cdot t_j}{\tau}\right)}{\sum_{m=1}^{N}\exp\left(\frac{t_i \cdot t_m}{\tau}\right)} \cdot \log\left(\frac{\exp\left(\frac{p_i \cdot t_j}{\tau}\right)}{\sum_{m=1}^{N}\exp\left(\frac{p_i \cdot t_m}{\tau}\right)}\right)\right]. \tag{3}$$

## 3 PARAMETERS COMPARISON

In the main paper, we mention that most multiple-subjects methods (such as Mindeye2 (Scotti et al., 2024b) and UMBRAE (Xia et al., 2024)) still require separate heads or tokenizers for each subject. As the number of subjects increases, their training costs and model parameters grow linearly, making this approach impractical for larger subject pools.

We use UMBRAE (Xia et al., 2024) as an example to show this linear growing of complexity. As shown in Tab. 6, each subject of UMBRAE (Xia et al., 2024) requires a tokenizer of 11.2M parameters. As the number of subjects increases, the number of parameters grows linearly to a very large size (*e.g.*, up to 11,310M with 1000 subjects). In contrast, the number of parameters of our method remains constant, regardless of the number of test subjects.

## 4 THE VISUALIZED IMAGE RETRIEVAL RESULTS

Fig. 1 depicts the image retrieval results of our visual brain decoding model, which is trained on 167 subjects from our consolidated HCP dataset. The results are extracted from the testing of Subject 1, which is an unseen subject to this model. We show the top three images that have the highest retrieval similarities to the target image seen by Subject 1. We can see from Fig. 1 that the retrieved images exhibit significant visual similarity to the target image. For example, in the first row, the target and retrieved images are all facial images, while the first retrieved image is exactly the target image. In the fourth row, all the retrieved images have two persons, as in the target image. In the last two rows, the retrieved images share similar colors and tones to the target images. Nonetheless, in some cases, such as the second retrieved image in the fifth row and the third retrieved image in the sixth row, the retrieval results are not satisfactory.

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

Figure 1: The visualized image retrieval results. 'Target images' refer to those viewed by Subj 1 from the HCP dataset, while 'Retrieved images' represent the corresponding retrieval outputs from the visual brain decoding model trained on 167 subjects. The retrieved images are ranked from left to right based on retrieval similarity.

et al. Mindeye2: Shared-subject models enable fmri-to-image with 1 hour of data. *arXiv preprint arXiv:2403.11207*, 2024b.

Weihao Xia, Raoul de Charette, Cengiz Öztireli, and Jing-Hao Xue. Umbrae: Unified multimodal decoding of brain signals. *arXiv preprint arXiv:2404.07202*, 2024.