# OpenReview forum: "Toward Generalizing Visual Brain Decoding to Unseen Subjects"
_ICLR.cc/2025/Conference — ICLR 2025 Poster_

### Official Review · Reviewer_cDXj · 2024-10-30

**Soundness:** 1
**Presentation:** 2
**Contribution:** 1
**Rating:** 5
**Confidence:** 4

**Summary:**

The paper aims to tackle the challenge of visual brain decoding with a focus on improving generalization across a broader population of subjects. The authors generate a new dataset based on the HCP dataset, where image-fMRI pairs are derived from the last frame of each second in a movie, with the fMRI signals averaged over a specific time window. To map brain signals to the CLIP embeddings of corresponding images, they employ a contrastive learning approach. The discrepancy in voxel counts among subjects is addressed through a straightforward upsampling technique. Furthermore, the authors propose a novel architecture designed to mitigate the exponentially increasing model size typically required by state-of-the-art methods when integrating additional subjects into the pipeline.

**Strengths:**

The paper aims to address two major challenges in the field of visual brain decoding: the scarcity of existing datasets and the potential scalability issues of current pipelines when expanding to a larger number of subjects. However, the evidence provided for the proposed method is insufficient to fully support its effectiveness in tackling these problems.

**Weaknesses:**

1) I have concerns regarding the dataset generation methodology. It appears that the fMRIs generated for each image may not accurately capture the brain's response to the specific features of those images. Specifically, the resulting fMRI signals seem to reflect an accumulation of activity from prior stimuli, rather than providing a clear, one-to-one representation of the neural response to individual image features. This raises questions about whether the fMRI data truly aligns with the visual stimuli as intended.

2) The validity of the generated dataset is questionable, particularly given the notable performance disparity observed between the NSD dataset and the paper's dataset (as noted in Table 2, row 1, compared to Table 6, row 2). It is unclear why there is such a significant difference, raising concerns about potential biases or inconsistencies in the data generation process that may not accurately reflect the neural representation of the images.

3) In line 485, the paper reports MindEye 1's top-1 accuracy as 84%. However, according to Scotti et al. (2024a), MindEye 1 achieved a top-1 accuracy of 93.1% for image retrieval and 90.1% for brain retrieval (Table 1, row 5). In contrast, Scotti et al. (2024b) present MindEye 1's results as an average across four individual models (Table 1, row 3). Therefore, despite the mention of "our implementation" in line 488 which is unclear, it appears that this approach does not surpass the current state-of-the-art performance.

4) Lines 51-52 suggest that hand-picking voxels is not ideal for data collection, yet the results in Table 6 (columns 5 and 7) demonstrate the superiority of this approach. This discrepancy raises questions about the claim. It is plausible that 'whole-brain' data could contain a significant amount of redundancy or noise, as suggested by prior research (Lin et al. "Redundancy and dependency in brain activities." Shared Visual Representations in Human & Machine Intelligence (2022).), which may explain the better performance of the hand-picked voxel strategy.

5) Table 6, row 7, indicates extremely poor performance, raising doubts about the pipeline’s ability to generalize effectively. In contrast, Scotti et al. (2024b), Table 1, row 7, report significantly better results even with a subset of the data. To strengthen the evaluation, a clear comparison between the proposed approach and MindEye2 is essential, as it would help clarify the relative effectiveness of the proposed method.

6) As the authors noted, previous studies have employed various methods to address the discrepancies in voxel counts across subjects, such as creating a common representational space (e.g., Scotti et al., 2024b). However, it is unclear how a simple upsampling approach effectively manages the differences in fMRI signals among subjects. More details are needed on the implementation of the upsampling process. Additionally, an in-depth analysis comparing upsampling to existing alignment methods (e.g., functional alignment, anatomical alignment, common representational space) is necessary to understand whether it can truly serve as a viable alternative.

**Questions:**

In addition to addressing the previously mentioned weaknesses, the following questions should be clarified to strengthen the paper:

Are there any validation checks that confirm whether the fMRI signals accurately reflect the intended neural responses to individual images?

Are there specific characteristics of whole-brain data that introduce redundancy or noise, which might explain the superior results of the hand-picked voxel approach?

In lines 66-67, it is stated that “the model becomes increasingly complex as the number of subjects increases, making it challenging to scale to a larger population,” referring to state-of-the-art architectures. It would be valuable to include a comparison of model size (in terms of the number of parameters) between the proposed approach and existing architectures. Such a comparison would help highlight the scalability advantages of the proposed model more clearly.

---

> ### Author Response · Authors · 2024-11-21
>
> We thank the reviewer for the detailed feedback on our paper. We would like to clarify that the purpose of our study is to explore methods to meet a major challenge — model’s generalization capabilities to unseen subjects.
>
> **1.Q: fMRIs for each image may not accurately capture the brain's response.**
>
> **R:** Accurately matching each video frame to brain response is nearly impossible. fMRI measures changes in the brain's blood oxygen levels, which do not correspond in a real-time one-to-one manner. However, there are many approved fMRI-video frame retrieval studies [1][2][3][4] which either use a delayed fMRI response (due to hemodynamic delay) or average fMRI voxels over a few seconds. These methods are widely accepted in the academic community, so we followed them [1]. Regarding the validation experiments, our stable training results across different subjects confirm the reliability of our data.
>
> **2.Q: The notable performance disparity observed between the NSD dataset and the paper's dataset. (Table 2, row 1, compared to Table 6, row 2)**
>
> **R:** This appears to be a misunderstanding. The large performance disparity likely arises from two factors: (1) The NSD experiments in Table 6 test on 300 images (see line 487), while the experiments in Table 2 retrieve from 100 images (line 350). This inherently leads to differences in performance. (2) The NSD dataset presents a more challenging task with more scenarios, so a lower retrieval performance is reasonable. This performance disparity does not indicate that our dataset is problematic.
>
> **3.Q: MindEye1's top-1 accuracy is mismatched, and our implementation is unclear.**
> **R:** This appears to be a misunderstanding. Our 84% accuracy aligns with the MindEye1 [5] training using CLIP loss (please see the second row of Table 3: 83.7% in the MindEye1 paper, also refer to the settings in line 346 of our paper). Additionally, when we reproduced MindEye1 with BiMixCo + SoftCLIP, the result was 93%, which is perfectly aligned with the original result. We do not report this in the paper, as our main experiments use CLIP loss. The reason for using CLIP loss in our experiments is that we want to verify a specific phenomenon, not to achieve SOTA results (see lines 463-465 of our paper).
>
> **4.Q: Hand-picking is not ideal for data collection, yet it is good for model performance.**
>
> **R:** We completely agree and acknowledge in lines 232-237 that hand-labeled data yields better performance, as it is directly linked to specific brain regions responsible for visual processing. Further, it is not ideal for research generalization. When working with hundreds or thousands of subjects, hand-labeling becomes impractical. Our approach prioritizes the scalability of data to explore generalization across a large number of subjects, rather than focusing solely on performance improvements. Therefore, hand-picking is not ideal for our work.
>
> **5.Q: Table 6, row 7, indicates extremely poor performance.**
>
> **R:** The poor performance here is expected. It reflects the challenge of generalizing when the number of subjects is small. The model was trained on Subj1, 3, 4, 6, 7 and tested on Subj2, 5, which demonstrates the difficulty of generalizing to unseen subjects with a limited number of training subjects.
>
> As discussed in Questions 3, 4, and 5, our research focuses on finding generalization capabilities to unseen subjects and exploring the possibility of training foundational large brain decoding models. Understanding the distinction between "seen" and "unseen" subjects is necessary. Previous work has typically focused on training and testing with seen subjects, unlike our setting. Prior to this work, researchers were unsure whether generalizing to unseen subjects was even possible. We do not focus on achieving SOTA results, as this is not the goal of our paper.
>
>
> —————————Because of the character number limit, we split the responses into two comments.——————————

---

> > ### Author Response · Authors · 2024-11-21
> >
> > —————————Because of the character number limit, we split the responses into two comments.—————————
> >
> >
> > **6. Q: The detail of upsampling, and the comparison with alignment methods.**
> >
> > **R:** The upsampling method used in our work is simple bicubic interpolation, which is only intended for size alignment. The specific processing and generalization capabilities depend on the network itself. We did not compare to alignment methods because such methods require information from aligned subjects (e.g., identical input-output pairs across subjects). Our approach, however, targets unseen subjects without any prior information, making traditional alignment methods unfeasible. If aligned subjects provide no information, there is no basis for alignment. Therefore, performance comparisons with alignment methods are not applicable. We explain this issue in more detail in our response to Reviewer x87x; please refer to it for further clarification.
> >
> > **7. Q: Model complexity comparison.**
> > **R:** Thank you for your suggestion. We will include a model complexity comparison:
> > | Number of Testing Subjects | Parameters of UMBRAE [6] |  Parameters of Ours (using the same backbone) |
> > |-----------|-----|------------|
> > | 1 | 112.63M | 112.63M |
> > | 10| 213.43M | 112.63M |
> > | 100 | 1221.42M | 112.63M |
> > | 1000 | 11310.43M | 112.63M |
> >
> > As shown in the table, in the UMBRAE [6] case, each subject requires a tokenizer of 11.2M parameters. As the number of subjects increases, the number of parameters grows linearly to a very large size. In contrast, the number of parameters in our method remains constant, regardless of the number of testing subjects.
> >
> > We hope these clarifications can help resolve some of misunderstandings, most of which stem from the fact that we are exploring a new and challenging problem. We will incorporate these details in the revised paper in response to this reviewer and other reviewers’ concerns.
> >
> > **References:**
> >
> > [1] In Medical Image Computing and Computer Assisted Intervention. (MICCAI 2020)
> >
> > [2] Semantic reconstruction of continuous language from non-invasive brain recordings. (Nature Neuroscience)
> >
> > [3] A Continuous Semantic Space Describes the Representation of Thousands of Object and Action Categories across the Human
> > Brain. (Neuron)
> >
> > [4] Mapping between fMRI responses to movies and their natural language annotations. (NeuroImage)
> >
> > [5] Reconstructing the mind’s eye: fmri-to-image with contrastive learning and diffusion priors. (NIPS)
> >
> > [6] UMBRAE: Unified Multimodal Brain Decoding. (ECCV)

---

> ### Author Response · Authors · 2024-12-01
> **A Kind Reminder To Participate In The Discussion**
>
> For all the concerns raised by Reviewer cDXj, we have provided detailed responses. Do the responses and results address your concerns adequately? Please note that the discussion period ends on Dec. 2. Kindly let us know if you have any unresolved issues or additional concerns. Thank you.

---

> > ### Comment · Reviewer_cDXj · 2024-12-02
> >
> > I sincerely thank the authors for their constructive feedback, which has provided significant clarification regarding their work and motivations. While their responses have addressed some of my initial concerns, a few critical issues remain. Specifically, I am still not fully convinced that the HCP dataset is entirely appropriate for this approach, nor do I believe that the fMRI data accurately and reliably represent the corresponding images. Overall, I feel that the contribution of the paper, while valuable, remains somewhat limited. That said, I acknowledge the effort and progress made by the authors in addressing these challenges and am inclined to slightly increase my score as a result.

---

> > > ### Author Response · Authors · 2024-12-02
> > >
> > > Many thanks for the reviewer's feedback. We are pleased that some misunderstandings have been clarified. Mentioned that "fMRI data can not accurately and reliably represent the corresponding images (video frames)" is a challenge faced by the entire field, not just this work. Addressing this issue is well beyond the scope of this paper. It is worth emphasizing that under such settings, a significant number of studies on fMRI-image (or video frame) retrieval, classification, and reconstruction are accepted by various journals and conferences every year. Therefore, despite certain limitations, we believe this approach remains academically acceptable.

---

### Official Review · Reviewer_QQAs · 2024-11-01

**Soundness:** 2
**Presentation:** 3
**Contribution:** 2
**Rating:** 6
**Confidence:** 4

**Summary:**

The paper explores the generalization properties of visual decoding models when trained and tested on different subjects. Different model architectures (3D ConvNet, 1D ConvNet, Transformer, MLP) were trained on subsets of a dataset containing fMRI brain activity data collected while subjects watched audio-visual movies. Models were trained to recover the seen video frame out of a batch of images. The use of an upsampling step made it possible to reuse a single model across all subjects (vs. subject-specific layer in previous work). Results show that model performance increases as more subjects are seen during training, and that distribution shift between training and test subjects can negatively affect performance.

**Strengths:**

* Originality: While the proposed methodology is similar to previous work, the paper studies a setting (generalization across multiple subjects) which has not been explored much before. The analysis of the impact of training data composition (gender bias and of train-test set similarity) is also interesting and original.
* Quality: The paper is of good quality and presents relevant experiments to support the claims made by the authors.
* Clarity: The paper is overall well and clearly written, and the problem setting is well introduced and supported.
* Significance: The results are promising and motivate further scaling up of brain activity datasets.

**Weaknesses:**

1. The description of the dataset curation and test set sampling procedure lacks details (see Q1-2). As movie frames are likely correlated and do not cover the same semantic space as “standard” brain-image datasets (in which the images are selected to cover varied categories, e.g. NSD), it is unclear whether the models might in fact overfit to the specific scenes seen in the training set. This could have an impact on the absolute performance values reported in the paper.
2. There is also missing information about the architectures (upsampling procedure, number of parameters, hyperparameter selection procedure; see Q4-5).

**Questions:**

1. The use of a video dataset to define an image retrieval task is interesting, however the distribution of images obtained by sampling frames every 1 s might be biased in a way that influences the interpretation of the results. Given frames within a scene are likely significantly correlated, the level of difficulty of the retrieval task on the test set will vary a lot if examples are strictly sampled from different scenes or not. Were the test images sampled uniformly across the 3127 images? A helpful visualization might be a grid of all sampled test images.
2. Similarly, if a frame $i_j$ from was selected to be part of the test set and the surrounding frames that are part of the same scene (e.g. $i_{j-1}$ and $i_{j+1}$, which could be very similar to $i_j$ if the visual stimulus changes slowly) end up in the training set, there is a strong potential for overfitting. Could this be the case here?
3. Averaging the 4-s window after the target frame is likely suboptimal as this misses the second half of the peak response (occuring at 4-s according to the reference cited). Have the authors looked at different pooling strategies (e.g. averaging seconds 2 to 6)?
4. How many parameters do the different model configurations have? How were the hyperparameters selected? For instance I find it surprising that such a large Transformer was used (24 layers according to Suppl. Figure 4).
5. What upsampling methodology was used?
6. Previous studies on the NSD dataset relied on precomputed beta values obtained through a GLM fitting procedure that help improve SNR. Can the authors confirm whether the “whole-brain” data they used is also betas or is it BOLD signals? This could also be a more likely explanation for the discrepancy between results on NSDGeneral and “Whole-brain” data in Table 6.
7. How were the images of Suppl Figure 1 selected? Are these retrieval results representative of the other images in the test set?
8. Other comments:
- At line 39, Benchetrit et al. 2023 is cited as an example of fMRI visual decoding, however they used MEG.
- In the last sentence of the conclusion, the authors probably meant to say “a brain decoding foundation model” (instead of “encoding”) as brain encoding would instead mean reconstructing brain activity from the stimuli.

---

> ### Author Response · Authors · 2024-11-21
>
> We thank the reviewer for the thoughtful review and the constructive feedback on our paper. We appreciate your recognition of our work. Your questions are all valuable. Please find our responses to the points you raised.
>
> **1.Q: Is the data uniformly sampled from the video?**
>
> **R:** The images are uniformly sampled, and we will consider some suitable visualization methods. We will release both the data and code.
>
> **2.Q: There is a risk of overfitting to video scenes?**
>
> **R:** The test images are uniformly sampled as well. As the reviewer noted, our videos contain only dozens of scenes and many similar frames, so it remains challenging for our models to generalize to more semantic scenarios. However, this possible overfitting does not undermine our conclusion that more subjects and greater similarity among subjects improve generalization capabilities.
> Below is the summary of an overfitting experiment about this concern:
>
> | Training Subjects | Testing Subjects |  TOP1 Accuracy |
> |-----------|-----|------------|
> | Subj1,2,5 | Subj5 | 100% |
> | Subj1,2,5 | Subj7 | 1% |
>
> In this experiment, we train models on 1,000 public images (seen by all subjects) from the NSD test set and test them on those same images. The models are completely overfitted to those images. As shown in the table, the model trained on Subj1,2,5 achieves nearly 100% accuracy on the seen subject Subj5, but performs poorly on the unseen subject Subj7 (1% accuracy). We believe that generalization occurs in two dimensions: subject generalization and semantic space generalization. NSD has a broad semantic space that supports semantic generalization, while our dataset contains a sufficient number of subjects to facilitate subject generalization. Thus, even if the semantic space is overfitted, it will not affect our conclusion that generalization across subjects is possible.
>
> **3.Q: Different averaging windows or other pooling strategies.**
>
> **R:** We have experimented with several pooling strategies (e.g., pooling next 2-6 voxels). While these strategies affect the absolute performance metrics to a small extent, they do not impact the generalization conclusions. Therefore, we adopted the current approach.
>
> **4.Q: Model parameters.**
>
> **R:** Currently, the networks are designed to align with the parameters of MindEye1 [1], which has around 900M parameters. However, our backbone is replaceable. We have also trained a 100M parameter transformer in previous experiments, and the generalization conclusions remain unchanged.
>
> **5.Q: Upsampling methodology.**
>
> **R:** The upsampling method we use is a simple bicubic interpolation, which does not require training. Its purpose is solely for size alignment, and the generalization capabilities depends on the network itself.
>
> **6.Q: Do you use betas or BOLD signals for NSD dataset?**
>
> **R:** We use beta values for both NSDGeneral and NSD “whole brain” data. NSDGeneral data are directly related to brain regions involved in visual processing, which generally results in better visual decoding performance (see line 232 in the paper). We also explain why we use “whole brain” data: although the decoding performance is not as high, it does not require manual labeling. Therefore, it can be scaled up to produce larger datasets, making it more practical and suitable for studying generalization.
>
> **7.Q: Demo images selection.**
>
> **R:** There are no quantifiable criteria for the demo image selection. We select a few from different scenes, showing both good and poor retrieval cases.
>
> We will provide a revised version and incorporate these points with more details as requested by the reviewer. We look forward to the reviewer’s continued positive rating of our paper.
>
>
> **References:**
>
> [1] Reconstructing the mind’s eye: fmri-to-image with contrastive learning and diffusion priors. (NIPS)

---

> > ### Comment · Reviewer_QQAs · 2024-11-25
> >
> > I thank the authors for their answers. As suggested, I believe answers to Q3-7 should be included in a revised version of the manuscript.
> >
> > As for Q1-2, the authors’ sole focus on “subject generalization” (vs. “semantic space generalization”) is now clearer. I understand there is no publicly available dataset that has both a high number of subjects like HCP and rich semantic coverage like NSD, therefore it is difficult to study both at the same time. However, I believe this is a limitation of the current sets of results, in that subject generalization is only shown to be possible when the semantic content of the test set overlaps fully (or close to fully) with the training set. Leaving out some scenes from the HCP-derived dataset to build a test set would allow a first exploration of this question.
> >
> > With this in mind, I maintain my previous score.

---

> > > ### Author Response · Authors · 2024-11-25
> > >
> > > Thank you for your understanding and positive feedback. The HCP dataset is too limited in semantic space to simultaneously demonstrate generalization in both semantic and subject dimensions. We also look forward to the availability of more comprehensive datasets, which would allow us to better explore this issue and even train general brain decoding models. We will include the various details mentioned in Q3-Q7 in the paper and appreciate your contribution to improving the quality of our manuscript.

---

### Official Review · Reviewer_x87x · 2024-11-03

**Soundness:** 3
**Presentation:** 3
**Contribution:** 2
**Rating:** 6
**Confidence:** 3

**Summary:**

The paper explores cross-subject visual brain decoding, specifically focusing on scenarios where no data from unseen subjects are included in the training model. The contributions are twofold: first, the authors propose an effective, straightforward approach to generate image-fMRI data at scale using the HCP video viewing dataset. Second, they introduce a decoding method based on CLIP encoding combined with cosine similarity decoding. Although the approach is admittedly simple, it is practical and well-justified.

**Strengths:**

The writing is clear, and the rationale is logical. The question of alignment across subjects, which the paper addresses, is timely and aligns with topics discussed at prominent venues.

**Weaknesses:**

The decoding methodology is likely the paper’s most relevant contribution to ICLR, though I have identified two significant limitations in this area.

Firstly, the authors show that incorporating data from additional subjects in training improves the model's performance, interpreting this as a solution to the alignment problem in neuroAI. In this field, we frequently encounter issues with requiring identical input-output pairs across subjects. While the authors suggest that their method mitigates this problem, they do not explain how. Is it possible that the model simply learns shared noise inherent to the videos across subjects as the data pool increases? We know that functional alignment remains a challenge because brain responses vary at the voxel level, even if the encoded information is similar across the whole region. It is unclear how, if at all, the proposed method addresses this challenge.

Secondly, the paper lacks a direct comparison with other state-of-the-art methods, which the authors partially review in the introduction. Could alternative methods achieve similar or even better performance? The authors suggest that other methods are computationally expensive in terms of parameters and training, yet they do not provide specific details. Quantitative comparisons and a more in-depth analysis of parameter costs are essential to substantiate this claim.

**Questions:**

Please provide a detailed rationale explaining why adding more subjects is beneficial. Specifically, how does this approach address the alignment problem? A clear explanation would help clarify the method's impact on the cross-subject variability and functional alignment issues inherent to neuroAI.

Please compare the performance of the proposed method with at least two other state-of-the-art approaches. This comparison would help establish its relative effectiveness and efficiency, particularly in terms of computational costs, accuracy, and robustness across subjects.

---

> ### Author Response · Authors · 2024-11-21
>
> We thank the reviewer for the thoughtful review and constructive feedback on our paper. We appreciate your recognition of our work, including the timely motivation and the practical method. Please find our responses to your points.
>
> **1.Q: How does this approach address the alignment problem?**
>
> **R:** The study is not designed to solve the alignment problem, and our approach does not involve alignment (except for data sizes alignment). The alignment problem typically refers to aligning a trained model to a specific subject (or vice versa). It requires identical input-output pairs across subjects, as the reviewer mentioned. However, our method focuses on testing subjects without any prior information. As for the source of generalization capabilities, we can describe it as "including" rather than "aligning". In other words, our method works well for unseen subjects because training samples may include the feature mapping patterns of unseen testing subjects. Experiments also support this explanation. When the number of subjects is the same, a more similar training group to a testing subject (such as sharing the same gender) will lead to better generalization. Moreover, when there are more training subjects, the generalization abilities becomes better, because more training subjects provide more possibilities to include the pattern of the unseen testing subjects. Our main contribution is to highlight this new approach: generalization abilities can be generated by “including” enough subjects without alignment (fine-tuning).
>
> **2.Q: Provide the comparison with the alignment methods.**
>
> **R:** As mentioned above, we do not directly compare to alignment or fine-tune methods because all such methods require information from aligned subjects (e.g., identical input-output pairs or training data from aligned subjects). This makes alignment inherently a "seen" problem. Our approach, however, is designed for zero-shot generalization, where no information is available about the unseen subjects. In this case, traditional alignment or fine-tune methods are not feasible. Therefore, we can not provide the performance comparison. However, we can provide a comparison in terms of model complexity:
> | Number of Testing Subjects | Parameters of UMBRAE [1] |  Parameters of Ours (using the same backbone) |
> |-----------|-----|------------|
> | 1 | 112.63M | 112.63M |
> | 10| 213.43M | 112.63M |
> | 100 | 1221.42M | 112.63M |
> | 1000 | 11310.43M | 112.63M |
>
> Many methods, such as UMBRAE[1], use different tokenizers for different subjects. As shown in the table, in UMBRAE’s case, each subject requires a tokenizer with 11.2M parameters. As the number of subjects increases, the parameters grow linearly, leading to a substantial increase in model size. In contrast, the parameters of our method remain constant regardless of the number of testing subjects.
>
> We hope these responses have addressed the reviewer’s concerns.
>
> **References:**
>
> [1] UMBRAE: Unified Multimodal Brain Decoding. (ECCV)

---

> > ### Comment · Reviewer_x87x · 2024-11-25
> >
> > While I am not entirely convinced by the authors' response, I have decided to revise my score for two reasons: (1) The concept of inclusion rather than alignment is intriguing. While I am confident it is not the sole solution for achieving out-of-distribution generalization (with the domain being the users), it likely constitutes an important component of the solution. (2) The discussion on model complexity is both compelling and thought-provoking. Overall, I believe this paper deserves to be presented at ICLR as it has the potential to contribute meaningfully to advancing the field.

---

> > > ### Author Response · Authors · 2024-11-25
> > >
> > > Thank you for your recognition and positive feedback on our work. We also believe that this is not the sole solution to the problem but does provide a new perspective. We will incorporate this discussion and explanation about "inclusion and alignment" into the paper. Thank you for your contribution to improving the quality of our manuscript.

---

### Official Review · Reviewer_DnEE · 2024-11-04

**Soundness:** 3
**Presentation:** 3
**Contribution:** 3
**Rating:** 6
**Confidence:** 4

**Summary:**

This paper presents a new brain decoding paradigm for zero-shot brain decoding. Multiple experimental design exposes that brain decoding models can be generalized from existing subjects to unseen subjects.The effect of inter-subject similarity, such as gender, on generalization ability is discussed for the first time.

**Strengths:**

1. This paper is well written and easy for the reader to read.
2. The motivation for the zero-shot setting is interesting.
3. The experimental design is very rich with multiple validations. Readers are inspired by explorations of the impact of gender.
4. In a sense, integrating the new dataset breaks the routine of NSD data use and is contributory.

**Weaknesses:**

1. From the point of view of machine learning, the algorithms for brain decoding in this paper are basically existing methods. The use of downsampling to obtain voxels of uniform size across multiple subjects has been used, e.g. CLIP-MUSED [1] uses PCA downsampling to align different subjects, also without additional training.
2. The methodology mentioned by the authors was not compared to any relevant studies quantitatively. The methods mentioned by the authors were not compared to any relevant studies, so it is difficult to comparatively measure whether the methods proposed by the authors are more effective or not. There are actually a subset of papers that have performed the appropriate anatomical alignment for zero-shot brain decoding[2, 3, 4]. Because of open source issues, a full reproduction may be a bit difficult, but small designs such as the use of **fsaverage** are easy to implement (fsaverage is already widely used in neuroscience).
3. While the new integrated dataset makes sense, it seems to me that a neuroscience research task without algorithmic design requires a dataset of its own design. I think this is missing in this paper. And the decoding task is very simple, containing only the retrieval task. Although the authors mention that the method can be generalized to the reconstruction task, a glance at related papers on the same method makes it hard for me to be convinced (the same method using downsampling fails miserably on the reconstruction task).

Overall I do not deny the contribution of this paper and I think the research ideas in this paper are very interesting and deserve to be published. The reason I am currently leaning towards rejection is that I am concerned that the paper may be more suited to brain science related journals or conferences than machine learning conferences like **ICLR**. Because the methods in this paper do not involve any representation learning methods or any algorithmic innovations, and the conclusions are more of a scientific experimental nature.

I would consider changing the rating to a positive one if the authors could convince me very well.

**Reference**
1. Zhou, Qiongyi, et al. "CLIP-MUSED: CLIP-Guided Multi-Subject Visual Neural Information Semantic Decoding." ICLR 2024.
2. Thual, Alexis, et al. "Aligning brain functions boosts the decoding of visual semantics in novel subjects." ArXiv 2023.
3. Bao, Guangyin, et al. "Wills Aligner: A Robust Multi-Subject Brain Representation Learner." Arxiv 2024.
4. Qian, Xuelin, et al. "fmri-pte: A large-scale fmri pretrained transformer encoder for multi-subject brain activity decoding." Arxiv 2023.

**Questions:**

I don't really agree with your statement in lines 506-508 of Section 4.6 that the poor generalization of the method on the NSD dataset is due to the problem of too few subjects. It seems to me at the moment that the integrated new dataset is consistent across all visual stimuli viewed by all subjects. So even for unseen subjects, the model is trained with the images that unseen subjects 'see'. However, for the NSD dataset, the images seen by different subjects, although they are all COCO images, are not the same in the training set. Therefore, in the absence of any comparison of related methods, how can it be proved that the method proposed in this paper is better not due to the difference in dataset ?

---

> ### Author Response · Authors · 2024-11-21
>
> We thank the reviewer for the thoughtful review and constructive feedback on our paper. We appreciate your recognition of our work, including the positive comment on the motivation and the completeness of the experiment and dataset. Please find our responses to your points.
>
> **1.Q: The Method is nothing new. Using downsampling.**
>
> **R:** We focused on the model’s generalization ability, proposing a novel unified framework with new learning paradigms to address a new problem, laying a foundation for future brain decoding models. These innovations pertain to both the methodology and the framework which are all related to the machine learning. Moreover, we employ an upsampling method instead of downsampling.
>
> **2.Q: There is no quantitative comparison to relevant studies.**
>
> **R:** The reviewer mentioned that alignment algorithms can not be directly compared to ours. These alignment algorithms require information from the testing subjects for fine-tuning (aligning) — such as anatomical data or a small set of new data from the testing subjects. Thus, these alignment methods are more akin to few-shot approaches rather than zero-shot methods. Our approach focuses on handling unseen subjects without any prior information about them. Our method only needs size alignment through upsampling. In this case, traditional alignment methods cannot be used. “fsaverage” is a useful feature alignment method and may enhance our approach. However, our method does not rely on alignment, even simple upsampling could achieve the necessary size adjustment effectively.
>
> **3.Q: A neuroscience research task requires a dataset of its own design. This paper deserves to be published but is more suited to brain science related journals.**
>
> **R:** Thanks again for your recognition. We believe ICLR is a very suitable conference. First of all, the scope of ICLR includes “applications to neuroscience & cognitive science”. Second, our paper is framed from the perspective of AI research, presenting a unified framework and learning paradigms. As the reviewer pointed out, we are not addressing a specific neuroscience problem; instead, we focus on identifying commonalities in brain activity patterns. Our ultimate goal is to develop a universal brain decoding model that can be applied to all subjects. Although we are still far from this goal, we are exploring its feasibility. In summary, we believe this paper fits ICLR well.
>
> **4.Q: Downsampling fails miserably in the reconstruction task.**
>
> **R:** Yes, we acknowledge that downsampling can be problematic. However, we use upsampling which retains all information without data loss. Our approach differs from Clip-Mused [1], which uses downsampling to reduce dimensionality (for classification). Our pipeline maps images and fMRI voxels to the CLIP feature space for retrieval, which is more similar to the Mindeye1 [2] framework. Therefore, we expect that our method should also perform well in reconstruction tasks, and this is a part of our ongoing work.
>
> **5.Q: Does poor generalization in the NSD dataset come from having too few subjects or because subjects see different images?**
>
> **R:** It is an insightful question. This experiment could help for clarifying:
> | Training Subjects | Testing Subjects |  TOP1 Accuracy |
> |-----------|-----|------------|
> | Subj1,2,5 | Subj5 | 100% |
> | Subj1,2,5 | Subj7 | 1% |
>
> We train the models on the 1,000 public images from the NSD test set and test them on those training images. In this case, every subject has seen the same images. As shown in the table, the model trained on Subj1,2,5 achieves nearly 100% accuracy on seen Subj5 but performs poorly on unseen Subj7 (1% accuracy). This suggests that the primary factor influencing generalization capabilities is the number of subjects, rather than whether subjects see the same images. Even when all subjects see the same images and testing occurs on the training set, a lack of subject diversity results in poor generalization capabilities to unseen subjects.
>
> We hope these responses address the reviewer’s concerns and that the reviewer is convinced to change the rating to a positive one.
>
> **References:**
>
> [1] CLIP-MUSED: CLIP-Guided Multi-Subject Visual Neural Information Semantic Decoding. (ICLR)
>
> [2] Reconstructing the mind’s eye: fmri-to-image with contrastive learning and diffusion priors. (NIPS)

---

> > ### Comment · Reviewer_DnEE · 2024-11-27
> > **Response to the Rebuttal**
> >
> > Many thanks to the authors for their replies, and part of my concern was solved. However, I am a little disappointed that the authors seem to have misunderstood other part of my comments, so I tend to maintain my rating. The following concerns remain unaddressed:
> >
> > **1.More about upsampling.** Thank you very much for pointing out the difference between upsampling and downsampling. When talking about multi-subject alignment, inconsistencies in voxel lengths are recognised that they need to be addressed. Therefore, whether using upsampling or downsampling is a detailed implementation, and the goal is the same. In this paper, the upsampling is the **core module** to achieve "To Unseen Subject" in a zero-shot setting, which authors emphasize that other method can not be used to. However, CLIP-MUSED's PCA downsampling and fsaverage both are pre-processes, which only needs operation in fMRI data without any training loss. My initial thought was that the authors could use these two methods as good baselines to highlight the advantages of their approach especially in the experiments, as you are currently lacking this comparison.
> >
> > **2. Method Comparasion.** As in Q1, the authors chose to ignore the correlation method and still believe they need additional alignment training, and I recommend that the authors read the relevant papers carefully. In fact, the up-sampling can't be said to be eye-opening, you can read Neuro-Vision[3] to understand what I'm saying, and this paper also has a pre-processing up-sampling of different subject fMRIs to the whole brain, and even takes spatial relationships as well as voxel activation states into account(The reason why I didn't mention it at first is that this article is the latest paper from this year's NeurIPS 2024).
> >
> > **3. Identifying Commonalities in Brain Activity Pattern.** On a methodological level, this paper does not seem to address the exploration of brain commonalities. When it comes to commonalities, the design of the methodology remains end-to-end, with up-sampling being just one of the implicit design steps. The authors did not do any representation experiments to confirm that brain commonalities were learned even partially, while I understand that this is difficult. By all accounts, I think the lack of upsampling comparison experiments is the biggest cause of this confusion.
> >
> > **4. Poor Generalization** Thanks to the authors for the impressive experiment on NSD. But I would like the authors to reveal the details of this experiment, did you test it on a retrieval pool with all training images? It seems to me that when only 1000 images are used for training, it's impossible to test with 100% accuracy on the remaining 26550 images.
> >
> > [1] Shen, Guobin, et al. "Neuro-Vision to Language: Image Reconstruction and Interaction via Non-invasive Brain Recordings." NeurIPS 2024

---

> > > ### Author Response · Authors · 2024-11-29
> > >
> > > We are particularly grateful to the reviewer for taking the time to continue this discussion with us. Your questions and concerns are invaluable in helping us clarify misunderstandings and improve the quality of our paper.
> > >
> > > We believe there might be a misunderstanding. Specifically, we wonder if you perceive the "upsampling" module as the key to generating generalization ability. We would like to clarify that. The upsampling module we use is a simple interpolation-based method that does not require training. It merely averages neighboring voxels to resize the input, which is even simpler than PCA or fsaverage (Upsampling doesn't even do feature extraction). Its sole purpose is to standardize input dimensions, enabling the model to handle data from unseen subjects. We would like to emphasize that in our framework, upsampling is not a critical component; it can be replaced by any size-unification method, such as PCA or fsaverage. Different size-unification only may affect downstream tasks (e.g., upsampling is more suitable for reconstruction than downsampling). Our reference to upsampling in the last response was only to address a typo (downsampling) in the comment. Importantly, it does not contribute to the model's generalization capability.
> > >
> > > The generalization ability in our work arises from two key factors:
> > >
> > > (1) A sufficiently large number of training subjects.
> > >
> > > (2) The presence of certain shared mapping patterns across subjects, which, to the best of our knowledge, has not been demonstrated in prior work.
> > >
> > > This generalization is not achieved through a powerful upsampling module or any other module. We believe this generalization stems from "including" rather than "aligning." In other words, our method performs well on unseen subjects because the training data likely encompasses the feature-mapping patterns of unseen testing subjects. This explanation is supported by our experiments. For example:
> > >
> > > (1) **Subject Similarity**: When the number of training subjects remains constant, a training group that is more similar to the testing subject (e.g., sharing the same gender) leads to better generalization.
> > >
> > > (2) **Increasing Training Data**: When the number of training subjects increases, generalization improves because the training set is more likely to include patterns representative of the unseen testing subjects.
> > >
> > > Our main contribution is highlighting this new perspective: generalization ability can arise from “including” enough subjects without requiring alignment (e.g., fine-tuning). (Reviewer x87x also expressed interest in it; please feel free to refer to the response for x87x.)

---

> ### Author Response · Authors · 2024-11-29
>
> After addressing this key misunderstanding, we now respond to the specific concerns (concerns 1-3 may stem from the above misunderstanding):
>
> **1. More about upsampling**:
>
> The purpose of upsampling is solely to enable the model to process data from unseen subjects, not to generate generalization abilities. As explained, this module is entirely replaceable and does not impact our conclusions about generalization. We do not consider this to be a novel module, nor do we believe it requires demonstrating its superiority (There's no superiority, just the simplest size alignment).
>
> **2. Method Comparison**:
>
> We now better understand the reviewer's concern about a lack of comparisons. When we stated "traditional alignment methods cannot be used," we referred specifically to functional alignment methods [1,2]. These methods cannot be generalized without alignment or fine-tuning. Of course, because size alignment is performed, they all can make inference on unseen subjects [1,2,3]. We are not claiming that their methods cannot perform inference on unseen subjects; rather, we are stating that without functional alignment, they lack generalization ability on unseen subjects (with only size-aligment and small number of training subjects, without functional alignment).
>
> In fact, our experiments (e.g., the last three rows in Table 6) reproduce and compare such scenarios. In these experiments, “ours” is just MindEye1 with size alignment, and it shows no generalization ability on unseen subjects (without fine-tuning on test subject data). Thus, our results demonstrate that these methods [1][2][3], while capable of performing inference on unseen subjects, do not achieve generalization without fine-tuning on unseen subjects.
>
> (Note: We carefully considered the references you provided, some of which we had previously reviewed. However, our discussion of “can be used” refers to different aspects (can perform inference VS have generalization ability). We summarize in the references section for clarity.)
>
> **3. Identifying commonalities in brain activity patterns**:
>
> It may be more precise to use the term "verify" here. The foundation of our work lies in commonalities in brain activity patterns. As mentioned earlier, generalization ability arises from these shared patterns, not from upsampling. These commonalities enable the training set to "include" unseen subjects, thereby producing generalization. While we have not localized these similarities to specific brain regions, we have at least verified their existence and demonstrated their role in driving generalization—an unexplored area in prior studies. For determining specific commonalities, as you mentioned, is extremely challenging and may reside in high-dimensional feature spaces rather than specific brain regions that are easily interpretable.
>
> **4. Poor Generalization**:
>
> It is expected that a model trained on 1,000 images cannot achieve good retrieval performance on the full NSD training set. This experiment was designed to show that even when the same images are seen, too few subjects fail to produce generalization. This supports our conclusion that generalization is primarily driven by the number of subjects, not shared images.
>
> Further, we believe generalization occurs along two dimensions: subject generalization and semantic space generalization. NSD offers a broad semantic space that supports semantic generalization, while our dataset provides sufficient subject diversity to support subject generalization. In other words, we can also say that NSD-based methods all have poor subject generalization due to their limited participant pool. Therefore, even if the semantic space is overfitted, our conclusion about cross-subject generalization remains valid. That is we mentioned, if there is a large dataset including enough subjects and image contents, we believe that it is possible to implement a general visual brain decoding model.

---

> ### Author Response · Authors · 2024-11-29
>
> **5. Innovations and contributions**:
>
> Finally, we would like to explain again the innovations and contributions of the article, which are not reflected in a certain module (such as upsampling). Our innovations lie in the whole pipeline we propose for studying generalization and advancing general brain decoding models. This includes dataset construction, learning paradigms, network architectures, and evaluation methods—all designed to expand applicability to more subjects and focus on generalization to new subjects.
>
> Using this pipeline, we demonstrate that generalization emerges from a sufficient number of subjects. This provides a new pathway: generalization does not require alignment (only needs simple size-unification). This exploration moves us closer to a general brain decoding model that can generalize directly to any subject (Not just can do inference , but have generalization abilities to all subjects).
>
> **A contribution example**: under our learning paradigm, exploring multi-subject generalization tasks with NSD is highly limited, as the dataset includes only 8 subjects. This severely limits generalization studies on new subjects. (Indeed, most of our early experiments with NSD yielded no generalization.) We do not advocate using NSD to study multi-subjects generalization ability, because even if there are some results, it will be difficult to verify on a larger number of subjects.
>
> We hope this response adequately addresses your concerns. Please feel free to reach out with further questions or feedback. Thank you again for your valuable comments, we will provide a discussion paragraph in the paper to make it clearer.
>
> **Reference and Summary**
>
> **[1] Aligning brain functions boosts the decoding of visual semantics in novel subjects**
>
> This paper proposes an alignment method. In the experiments, they explicitly mention that “with just 30 minutes of data, left-out subjects can reach performance which would have needed roughly 100 minutes of data in a within-subject setting.” This clearly indicates that they use new subject data for alignment, which is essential to their approach. Therefore, we state that a direct comparison with their alignment approach is not possible due to the fundamentally different setups.
>
> **[2] Wills Aligner: A Robust Multi-Subject Brain Representation Learner**
>
> This paper introduces a method for learning effective shared representations across multiple subjects. Their experiments involve training on multiple subjects and testing within the training set subjects. When testing on unseen subjects, they rely on a few-shot mapping process, as noted in Tables 4 and 5, which explicitly list the proportion of few-shot data used. If there were a row in their results with a few-shot proportion of zero, it would correspond to our experimental setup. However, since their approach depends on using some data from new subjects, it fundamentally differs from ours, where no new subject data are used for alignment or mapping.
>
> **[3] Neuro-Vision to Language: Image Reconstruction and Interaction via Non-invasive Brain Recordings**
>
> This paper also does training on multiple subjects and testing within the training subjects. It does not address generalization to unseen subjects.
>
> **[4] fMRI-PTE: A Large-scale fMRI Pretrained Transformer Encoder for Multi-Subject Brain Activity Decoding**
>
> We find the results of this paper puzzling. It utilizes fMRI reconstruction as a proxy task for self-supervised pretraining. According to Figure 2, the pretraining requires fine-tuning when applied to downstream tasks, similar to MAE. However, their description of the fine-tuning process is just a single sentence (see Page 8, Section 4.2.2). Furthermore, neither the model, code, nor supplementary materials are publicly available. During their pretraining on the UKB dataset, the model did not see any image data. After an unspecified fine-tuning process without any description, the model is claimed to reconstruct images on unseen NSD subjects. Is that reasonable? Without more details about their fine-tuning process, we can not include this paper's results as a valid reference.

---

> ### Comment · Reviewer_DnEE · 2024-12-01
> **Official Response by Reviewer**
>
> Many thanks to the authors for their efforts on rebuttal, and for two rounds of detailed and nuanced response attitudes. This reinforced my understanding of this paper and corrected some misconceptions. I started this paper on a positive note, believing that only a few concerns needed to be clarified. It was not well resolved due to some misunderstandings that arose from the first round of discussions. Now I think the authors have done a good job of responding accordingly.
>
> Although I still think that the addition of a certain number of comparison experiments could have been more enlightening for the researchers involved, as how to learn commonalities between subjects while avoiding fMRI noise is what the up-sampling module needs to focus on. However, this does not detract from the contribution of this paper, so I have increased my rating. Good luck to all of you!

---

> > ### Author Response · Authors · 2024-12-01
> >
> > We sincerely thank the reviewer for participating in the discussion and providing positive feedback. We greatly appreciate the professionalism demonstrated during the discussion. In our revisions, we will include clearer comparative experiments and explanations to minimize potential misunderstandings. As you noted, we adopted the simplest upsampling to demonstrate the general applicability of the conclusions. Moving forward, exploring better ways to extract shared features and reduce noise to enhance generalization capabilities will also be a priority. Thank you again for your time and effort!

---

### Author Response · Authors · 2024-12-02
**A Thanks Letter and Summary**

We thank all reviewers and area chairs for their valuable time, feedback, and thoughtful comments. Some comments provides valuable suggestions for improving the quality of our paper and guiding future research directions. As the ICLR reviewer-author discussion period comes to a close, we would like to provide a brief summary of the discussions.

This paper explores a novel challenge: generalizing visual brain decoding models to unseen subjects. Following the discussions, Reviewers DnEE, x87x, and QQAs have expressed positive opinions about our work, acknowledging its clear writing, detailed experiments and reasonable analyses. Reviewer DnEE remarked, “The motivation for the zero-shot setting is interesting”. Reviewer QQAs highlighted that “The analysis of the impact of similarity is interesting and original”. Reviewer x87x noted, “It has the potential to contribute meaningfully to advancing the field”. These comments reflect their support for the paper’s acceptance.

In contrast, Reviewer cDXj expressed a strong reject. We believe this stems from misunderstandings regarding both the purpose and the experimental setup of our work. To address these, we provided detailed explanations during the discussion phase. Unfortunately, Reviewer cDXj has not participated in any discussions, leaving us uncertain as to whether their concerns have been resolved.

Once again, we sincerely thank all reviewers and area chairs for their time, effort, and valuable feedback.

Sincerely, Authors of #1426

---

> ### Author Response · Authors · 2024-12-04
> **Update the Summary**
>
> Since Reviewer cDXj participated in the final discussion, I will update the summary accordingly. We addressed most of Reviewer cDXj’s concerns. The remaining concern revolves around: "I am still not fully convinced that the HCP dataset is entirely appropriate for this approach, nor do I believe that the fMRI data accurately and reliably represent the corresponding images."
>
> We would like to emphasize that, this usage of the HCP dataset represents one of the best available choices for exploring generalization capabilities now, because of its sufficient number of participants. Additionally, "fMRI data cannot accurately and reliably represent the corresponding images (video frames)" is a challenge faced by the entire field, not just this work. Under such conditions, a significant number of studies on fMRI-image (or video frame) retrieval, classification, and reconstruction papers are accepted by various journals and conferences each year. Therefore, despite certain limitations, we believe this approach remains academically acceptable (The other three reviewers also accept this approach).
>
> Finally, we sincerely thank all reviewers and area chairs for their time, effort, and valuable feedback.
>
> Sincerely, Authors of #1426

---

### Meta-Review · Area_Chair_Ct8d · 2024-12-19

**Metareview:**

This submission contributes a method for brain-based decoding generalizing across subjects. The study is a vision task, mapping brain signals to CLIP embeddings of the images. The submission generated interest and discussion from the reviewers, in particular the across-subject generalization. The reviewers appreciated the empirical design, however the comparison to baselines is not clear at all, which is very unfortunate. The other should strive to address this last point, clarifying the corresponding points raised in the discussion and ideally running experiments to complement them.

**Additional Comments On Reviewer Discussion:**

There was a good discussion with much back and forth between authors and reviewers. The discussion led to improving the manuscript.

---

### Decision · Program_Chairs · 2025-01-22

Accept (Poster)